# Enantioselective nickel-catalyzed anodic oxidative dienylation and allylation reactions

Qinglin Zhang[1], Jiayin Zhang[1], Wangjie Zhu[1], Ruimin Lu[1] & Chang Guo [1] ✉

Precision control of stereochemistry in radical reactions remains a formidable challenge due to the prevalence of incidental racemic background reactions resulting from undirected substrate oxidation in the absence of chiral induction. In this study, we devised an thoughtful approach−electricity-driven asymmetric Lewis acid catalysis−to circumvent this impediment. This methodology facilitates both asymmetric dienylation and allylation reactions, resulting in the formation of all-carbon quaternary stereocenters and demonstrating significant potential in the modular synthesis of functional and chiral benzoxazole-oxazoline (Boox) ligands. Notably, the involvement of chiral Lewis acids in both the electrochemical activation and stereoselectivity-defining radical stages offers innovative departures for designing single electron transfer-based reactions, significantly underscoring the relevance of this approach as a multifaceted and universally applicable strategy for various fields of study, including electrosynthesis, organic chemistry, and drug discovery.

The contemporary practice of stereoselective cross-coupling has emerged as a potent method for constructing a diverse range of chiral scaffolds with manifold applications from drug manufacture to materials engineering[1,2]. Among a plethora of possible activations, chiral Lewis acid catalysts have sublimely facilitated anionic strategies to dictate absolute stereochemistry with carbonyl compounds (Fig. 1a, top)[3–14]. Meanwhile, electrochemistry has emerged as a promising, sustainable, and ecologically benign tool for redox transformations that can procure direct addition or removal of electrons between the substrate and the electrode through a single electron transfer (SET) pathway. Merging electrochemistry with asymmetric Lewis acid catalysis could present an approach to engender α-carbonyl radical species from enolate intermediates (Fig. 1a, bottom)[15–18]. The extraordinary reactivity of chiral catalyst-bound open-shell intermediates can significantly enhance the efficacy of C-C bond-forming events for the construction of quaternary stereocenters with nucleophilic organosilanes, which would be challenging to achieve through conventional ionic transformations[19].

The pursuit of highly enantioselective electrochemistry-driven asymmetric catalysis is indeed a formidable task, mainly due to the nonspecific nature of radical intermediates[20–22]. The key challenge lies in the search for chiral catalytic systems that can operate effectively under electrochemical conditions while exhibiting high levels of stereocontrol[23–38]. In radical-involved electrochemical oxidative cross-coupling reactions, the direct oxidation of an achiral substrate in the absence of a chiral catalyst would result in the formation of racemic products, and this uncontrolled background process would significantly impede the attainment of precise and sophisticated stereo-selectivity, acting as a major barrier in this field[19] (Fig. 1b, path I). The development of a generally applicable strategy for eliminating uncatalyzed background reactions while ensuring that the catalytic asymmetric pathway exceeds a certain threshold is crucial in the discovery of efficient enantioselective oxidative radical cross-coupling reactions.

After conducting a comprehensive assessment, we formulated a hypothesis indicating that Lewis acid-catalyzed electrolysis would represent an exceptionally effective approach to achieve stereo-chemistry in radical-mediated cross-coupling reactions (Fig. 1b, path II)[39–43]. Recently, Meggers[39,40] and our group[41–43] showed the use of Lewis acid catalysis in asymmetric radical electrochemical transformations of 2-acyl imidazoles. This strategy possesses a host of distinctive features that deftly suppress racemic background reactions, presenting an auspicious avenue for augmenting oxidative cross-

[1]Hefei National Research Center for Physical Sciences at the Microscale and Department of Chemistry, University of Science and Technology of China, Hefei, China. ✉e-mail: guochang@ustc.edu.cn

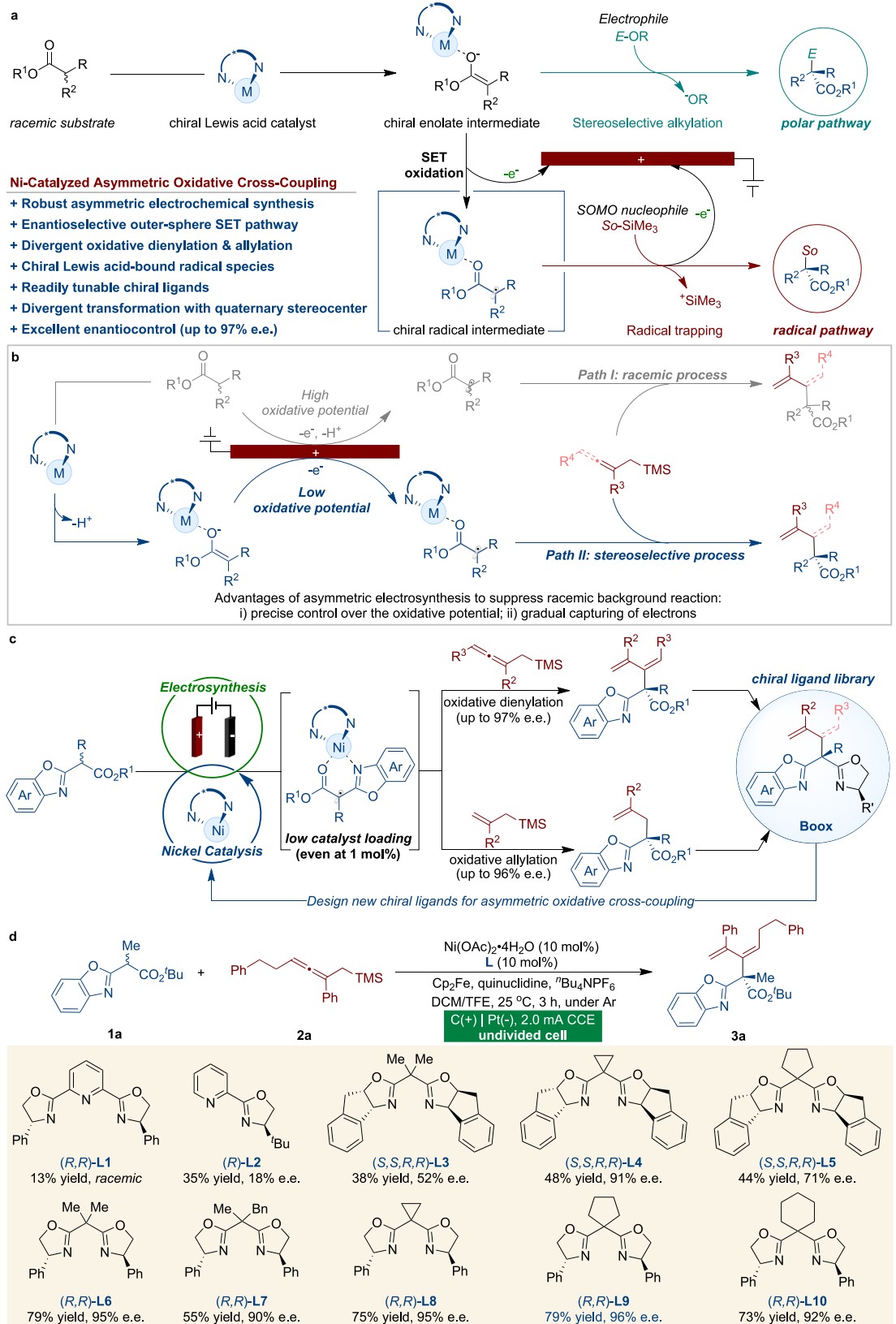

**Fig. 1 | Design of enantioselective anodic oxidative dienylation and allylation.** **a** Lewis acid-catalyzed anionic pathway and asymmetric electrolysis. **b** Advantages of asymmetric electrolysis. **c** Enantioselective nickel-catalyzed oxidative cross-coupling reactions. **d** Optimization of the reaction conditions. Reactions were carried out with **1a** (0.1 mmol), **2a** (0.3 mmol), $^nBu_4NPF_6$ (0.15 mmol), quinuclidine (0.05 mmol), Ni(OAc)$_2$·4H$_2$O (10 mol%), **L** (10 mol%), Cp$_2$Fe (10 mol%), and DCM/TFE = 2:1 (3 mL) at 25 °C in an undivided cell, 3 h, E/Z > 20:1. TFE, 2,2,2-trifluoroethanol.

coupling through superior enantioselectivity. Primarily, the coordination of the catalyst with the substrate results in a decreased oxidation potential, facilitating the electrocatalytic process and effectively reducing the likelihood of competing racemic reactions. Second, the easily tunable anode redox potential and current density enable precise control over the selectivity and rate of electron transfer, and further facilitate the selective activation of the catalyst-bound substrate while keeping the free substrate inert. This capability confers a distinct advantage to asymmetric electrochemical reactions compared to conventional processes that rely on stoichiometric chemical oxidizing reagents. Third, the structurally customizable Lewis acid complexes offer a pliable chiral cavity that effortlessly binds to both the carbonyl substrate and in situ generated radical intermediates, guiding the resultant cross-coupling reaction toward maximal efficacy asymmetric induction. Notably, the ability to achieve a stereoselective radical attack on an allene moiety poses a difficult yet potent synthetic challenge[44,45]. In the present study, we showcase the utility of electricity-driven asymmetric catalysis in highly selective dienylation and allylation reactions under mild anodic oxidation conditions (Fig. 1c), resulting in good yields, an extensive substrate scope, and remarkable enantioselectivities (up to 97% e.e.). Importantly, the utilization of a Lewis acid-catalyzed electrochemical platform presents innovative opportunities to craft robust enantioselective radical transformations.

## Results

### Design of the asymmetric anodic radical dienylation

To validate the feasibility of our hypothesis, we initiated reaction development with racemic benzoxazolyl acetate (**1a**) and allenylmethylsilane (**2a**) in the presence of a nickel catalyst at room temperature (Fig. 1d). The reaction was performed in an undivided cell outfitted with a graphite (C) anode and a platinum (Pt) cathode, utilizing constant current electrolysis (CCE) to identify a suitable chiral ligand. Desired product **3a** was produced, albeit in low yield and poor enantiomeric excess, using Ni(OAc)₂·4H₂O as a Lewis acid in conjunction with either a tridentate pyridine bisoxazoline (PyBox) ligand (**L1**) or bidentate pyridine oxazoline (**L2**). Remarkably, the indane-fused Box ligand (**L3**) provided a higher degree of enantioselectivity (52% e.e.). Further evaluation was conducted on a range of nickel complexes bearing different substituents on indane-fused Box ligands (**L4** and **L5**). The chiral ligand **L4**, with a cyclopropyl moiety, led to a significantly improved enantioselectivity (91% e.e.). Additionally, Box ligands **L6-L10**, which bear phenyl groups on the oxazoline rings, were investigated. Ligand **L9**, with a cyclopentyl moiety, was shown to be the optimal ligand, providing desired product **3a** in 79% yield and with 96% e.e.

### Mechanistic investigation

To elucidate the mechanism underlying the mild anodic oxidative coupling reactions with high levels of stereocontrol, a series of experiments were performed. Mechanistic studies were initiated by employing cyclic voltammetry (CV) to investigate the reaction components. Cyclic voltammograms revealed that the onset oxidation potentials for **1a** and **2a** were approximately +1.98 V and +1.11 V versus a saturated calomel electrode (SCE), respectively (Fig. 2a). Notably, the onset oxidation potential of **1a** with the nickel complex and quinuclidine was observed to be significantly reduced (Fig. 2b, +0.36 V versus SCE), indicating a lower energy requirement for oxidation. This shift in potential suggests the formation of a catalyst-bound reactive intermediate, which plays a crucial role in facilitating the oxidation process. Conversely, there was no appreciable alteration in the redox behavior of **2a** in the presence of the nickel catalyst (Fig. 2c). In a current-switched on-off experiment, the formation of coupling product **3a** was observed exclusively when the experimental system was subjected to electric conditions (Fig. 2d). To explore the kinetics governing the

process, we conducted a thorough examination of the reaction (Fig. 2e). Intriguingly, the anodic dienylation process can proceed without the presence of a nickel catalyst. The catalytic efficiency of this process was improved, albeit not much, by the addition of a chiral nickel catalyst. Consequently, it may seem paradoxical to achieve high enantioselectivity in the presence of a significant background reaction, as the uncatalyzed reaction would typically lead to the formation of a racemic product. Under electrolytic conditions, the concentration of the nickel catalyst with chirality was found to impact the yield of adduct **3a** (Fig. 2f, blue line). However, the enantiomeric excess of the product (Fig. 2f, green line) was not noticeably influenced. Catalyst loadings varying from 0.5 mol% to 10 mol% produced coupling adduct **3a** with a similar enantiomeric excess in each case (Fig. 2f, green line), indicating that the racemic background reaction contributes minimally to the observed stereoselectivity. The chiral nickel catalyst, even at low loading (0.5 mol%), effectively overrides the background reaction, allowing for the desired enantioselective outcome to be achieved. In contrast, the utilization of silver oxide as a chemical oxidant instead of an electrical input revealed a noticeable decline in the enantiomeric excess of the coupling adduct **3a** as the catalyst loading decreased from 10 mol% to 0.5 mol% (Fig. 2f, red line). Significantly, the use of a 0.5 mol% catalyst loading led to a yield of 30% and an enantiomeric excess of only 11%, which further highlights the superiority of asymmetric electrochemical systems. Compared to conventional processes that rely on stoichiometric chemical oxidants, our anodic oxidative coupling methodology provides distinct advantages through its ability to tune oxidation potential and current density to achieve controllable oxidation processes, which are slower than the rate of the nickel-catalyzed reaction. Furthermore, the catalytically generated intermediates are inherently more susceptible to oxidation than the reactants (Fig. 2b, green line), also playing a crucial role in the suppression of racemic background reactions.

To better comprehend the underlying mechanistic details of the process, a series of control experiments were conducted (Fig. 2g). When the reaction was carried out in the absence of ferrocene (Cp₂Fe, entry 2, 27% yield, 96% e.e.) or under air conditions (entry 3, 38% yield, 95% e.e.), significantly lower yields were obtained. The absence of a chiral catalyst did not significantly hinder the progression of the reaction, as a comparable yield of the racemic product was obtained (entry 4), suggesting the involvement of a strong background process in this reaction. In the absence of an electric current source, the process was entirely inhibited (entry 5), which is consistent with our results shown in a current-switched on-off experiment (Fig. 2d). To further confirm the pathway of the reaction, we conducted a controlled potential experiment with a constant anodic potential of +0.4 V (versus SCE, Fig. 2b) (entry 6), a potential insufficient for the direct oxidation of allenylmethylsilane **2a** ($E_{onset} \approx$ +1.11 V versus SCE, Fig. 2c). The desired product **3a** was obtained in 75% yield and 96% e.e., indicating that anodic oxidation of **2a** is unlikely to play a substantial role in the observed asymmetric dienylation reaction. Additionally, when allylsilane with a cyclopropyl moiety **4** was subjected to electrolytic reaction conditions, ring-opening products **5** or **6** were obtained, further supporting the formation of the α-carbonyl radical intermediate in the enantioselective anodic dienylation protocol (Fig. 2h).

Based on the results of our mechanistic investigations, we propose a plausible catalytic mechanism for electrically triggered asymmetric catalysis. As depicted in Fig. 2i, the catalytic cycle begins with the interaction of **1** with the nickel catalyst to ultimately form nickel-bound enolate intermediate **I**, which is preferentially oxidized over the catalyst or substrates. A ferrocene-assisted anodic oxidation of **I** through a single-electron transfer pathway results in the creation of a nickel-coordinated radical intermediate **II**, which is subsequently captured by allenylsilane to establish a stereogenic center within intermediate **III**. DFT calculations suggested that radical intermediate **III** is oxidized to a cation **IV** and then undergoes TMS removal to afford

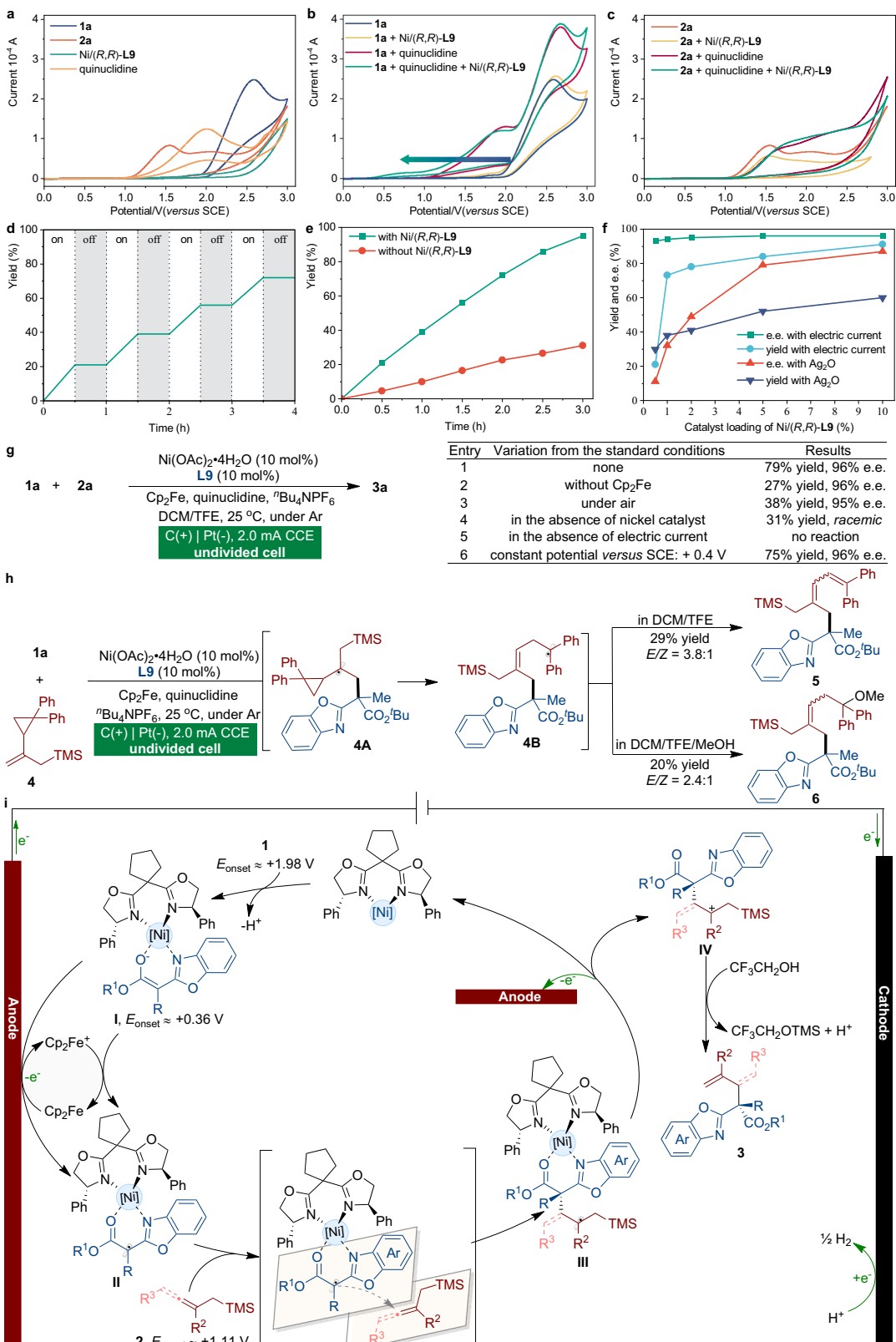

**Fig. 2 | Mechanistic investigations. a** CV of related compounds in the catalytic system. **b** CV of **1a** in the absence or in the presence of a Lewis acid catalyst. **c** CV of **2a** in the absence or in the presence of a Lewis acid catalyst. **d** Profile of a current-switched on-off experiment. **e** Kinetic studies to investigate the racemic background reaction. **f** Investigation of the catalyst loading. The reaction condition with silver oxide as a chemical oxidant instead of an electrical input: reactions were carried out using racemic benzoxazolyl acetate **1a** (0.1 mmol), allenylmethylsilane **2a** (0.30 mmol), $^nBu_4NPF_6$ (0.15 mmol), quinuclidine (0.05 mmol), Ni(OAc)$_2$·4H$_2$O/ **L9** (0.5–10 mol%), Cp$_2$Fe (0.5-10 mol%), Ag$_2$O (0.30 mmol), and DCM/TFE = 2:1 (3 mL) at 25 °C. **g** Control experiments. **h** Radical clock experiment. **i** Proposed catalytic cycle.

## Table 1 | Enantioselective anodic oxidative dienylation with allenylsilanes

Reaction scheme: compound **1** + compound **2**, with conditions Ni(OAc)$_2$·4H$_2$O (10 mol%), **L9** (10 mol%), Cp$_2$Fe, quinuclidine, $^n$Bu$_4$NPF$_6$, DCM/TFE, 25 °C, 3 h, under Ar. C(+) | Pt(−), 2.0 mA CCE, undivided cell → compound **3**

Scope of allenylsilanes

**3a** 79% yield, 96% e.e.
**3b** 69% yield, 95% e.e.
**3c** 71% yield, 96% e.e.
**3d** 69% yield, 96% e.e.
**3e** 60% yield, 97% e.e.

**3f** 85% yield, 96% e.e.; 86% yield, 96% e.e.[a]
**3f** CCDC (**2268498**)
**3g**, X = F, 68% yield, 94% e.e.[b]; **3h**, X = Me, 62% yield, 96% e.e.[b]
**3i**, X = Cl, 66% yield, 95% e.e.[b]; **3j**, X = Me, 79% yield, 95% e.e.[b]
**3k** 75% yield, 97% e.e.

Scope of benzoxazolyl acetates

**3l** 52% yield, 97% e.e.[c]
**3m** 67% yield, 96% e.e.[d]
**3n** 63% yield, 95% e.e.[b,d]
**3o** 71% yield, 95% e.e.[b,d]
**3p** 72% yield, 95% e.e.[b,d,f]

**3q** 64% yield, 96% e.e.[d]
**3r** 63% yield, 92% e.e.[d]
**3s** 61% yield, 88% e.e.[b]
**3t** 64% yield, 86% e.e.[b,e]
**3u** 61% yield, 90% e.e.

**3v** 61% yield, 90% e.e.
**3w** 61% yield, 85% e.e.[b,e]
**3x** 78% yield, 89% e.e.[b,e]
**3y** 62% yield, 92% e.e.
**3z** 59% yield, 78% e.e.[g]

**3aa** 41% yield, 94% e.e.[g]
**3ab** 51% yield, 86% e.e.
**3ac** 54% yield, 88% e.e.
**3ad** 56% yield, 89% e.e.
**3ae** 62% yield, 98% e.e.

Unless otherwise specified, all reactions were carried out using racemic benzoxazolyl acetate **1** (0.1 mmol), allenylmethylsilane **2** (0.30 mmol), $^n$Bu$_4$NPF$_6$ (0.15 mmol), quinuclidine (0.05 mmol), Ni(OAc)$_2$·4H$_2$O (10 mol%), **L9** (10 mol%), Cp$_2$Fe (10 mol%), and DCM/TFE = 2:1 (3 mL) at 25 °C for 3 h under constant-current conditions in an undivided cell. [a]0.5 mmol scale, 3.5 h. [b]HCO$_2$H (20 mol%). [c]DCE/TFE (3 mL, 2:1) at 40 °C. [d]**L4** (10 mol%). [e]Allenylmethylsilane **2** (0.60 mmol). [f]6 h. [g]Ni(OAc)$_2$·4H$_2$O (20 mol%), **L9** (20 mol%), **2** (0.6 mmol) at 50 °C.

the corresponding product **3** (see Supplementary Information for details).

### Nickel-catalyzed enantioselective anodic oxidative dienylation

With these optimized reaction conditions for enantioselective anodic oxidative dienylation, we proceeded to examine the versatility of this reaction through the use of various substituted allenylsilanes **2**. As exemplified in Table 1, diverse alkyl-substituted allenylsilanes **2** proved to be excellent reaction counterparts, furnishing the expected adducts in high yields and remarkable enantioselectivities (**3a**–**3f**). The absolute configuration of **3f** was accurately determined by X-ray diffraction analysis, while the conformations of the other products were assigned

## Table 2 | Enantioselective anodic oxidative allylation reactions

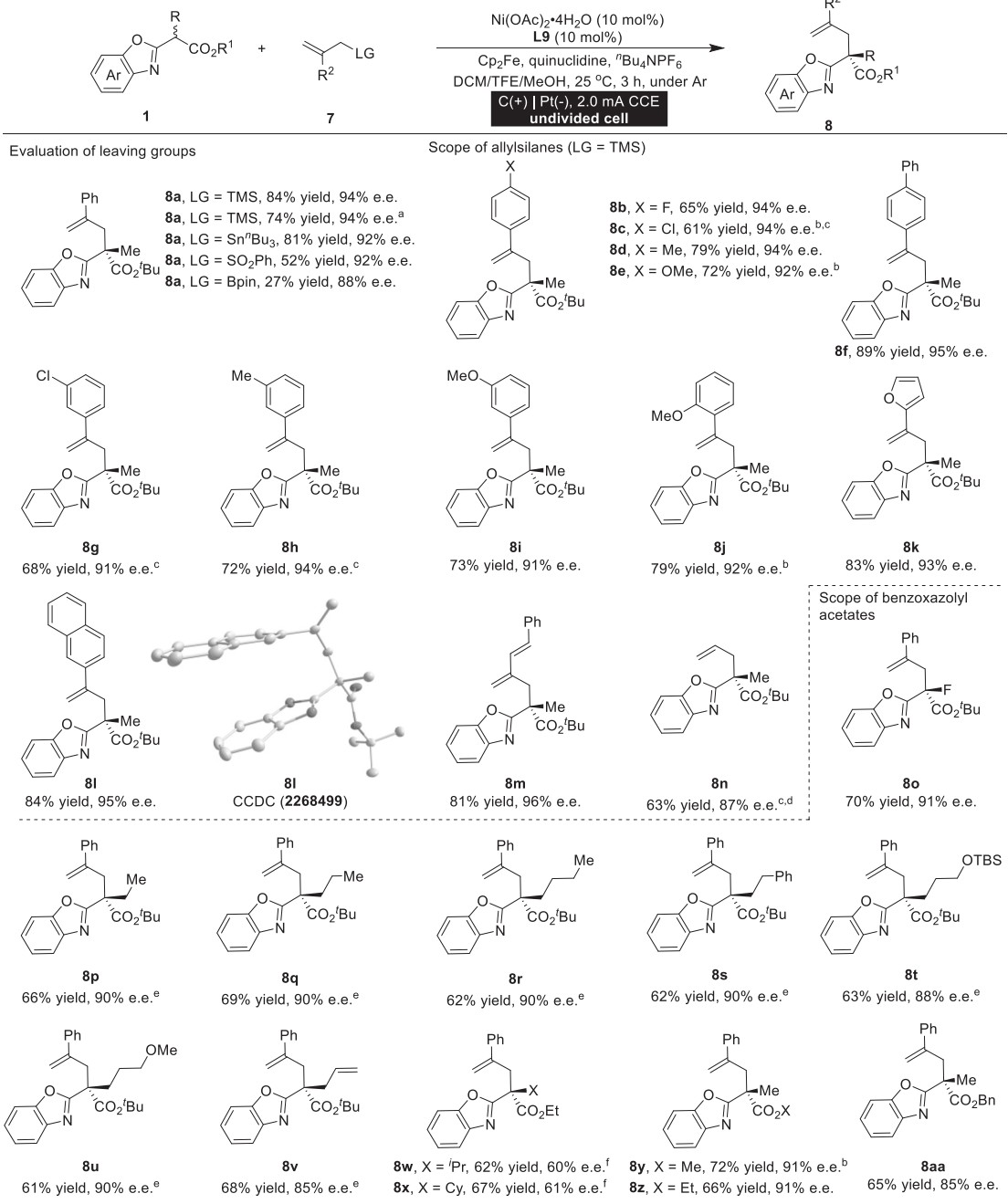

Unless otherwise specified, all reactions were carried out using racemic benzoxazolyl acetate **1** (0.1 mmol), allylic counterpart **7** (0.30 mmol), $^nBu_4NPF_6$ (0.15 mmol), quinuclidine (0.05 mmol), Ni(OAc)$_2$·4H$_2$O (10 mol%), **L9** (10 mol%), Cp$_2$Fe (10 mol%), and DCM/TFE/MeOH = 3:2:1 (3 mL) at 25 °C for 3 h under constant-current conditions in an undivided cell. [a]0.5 mmol scale. [b]HCO$_2$H (20 mol %). [c]6 h. [d]Potassium allyltrifluoroborate (0.30 mmol, 3.0 equiv) and HCO$_2$H (10 mol%) were used, 25 °C, 6 h. [e]10 °C. [f]With **7** (0.6 mmol) at 50 °C.

by analogy. Remarkably, various substitution patterns at the aromatic moiety of allenylsilanes **2** were well tolerated, regardless of their underlying electronic and steric properties (**3g**–**3k**). Furthermore, this methodology exhibited compatibility with allenylsilane possessing dialkyl substitution, and the desired product was obtained in 52% yield and 97% e.e. (**3 l**). The scope of the transformation was further extended through an examination of substituted benzoxazolyl acetates **1**. A wide range of **1** with different α-substituents, including alkyl (**3m**–**3q**) and allylic (**3r**) groups could be well tolerated, and good chemical yields and enantioselectivities were observed. Variation of the ester group of benzoxazolyl acetates **1** had a certain impact on the reaction outcomes (**3 s** and **3t**). Diverse substitutions of benzoxazolyl acetates

at the aromatic ring exerted only a minimal impact on the reaction while ensuring good levels of enantioselectivity (**3u**–**3y**). Varying the substrate **1** with ketone (**3z**) and amide (**3aa**) gave moderate yields. The ester size was varied (**3ab**-**3ae**), and employing sterically demanding benzoxazolyl acetate as the substrate resulted in enhanced enantioselectivity of the reaction (**3ae**).

### Nickel-catalyzed enantioselective anodic oxidative allylation

To further demonstrate the versatility of this nickel-catalyzed electrochemical reaction, we carried out an asymmetric anodic oxidative allylation reaction of racemic benzoxazolyl acetates **1** and allylic counterparts **7** under optimized conditions (Table 2). Gratifyingly, the

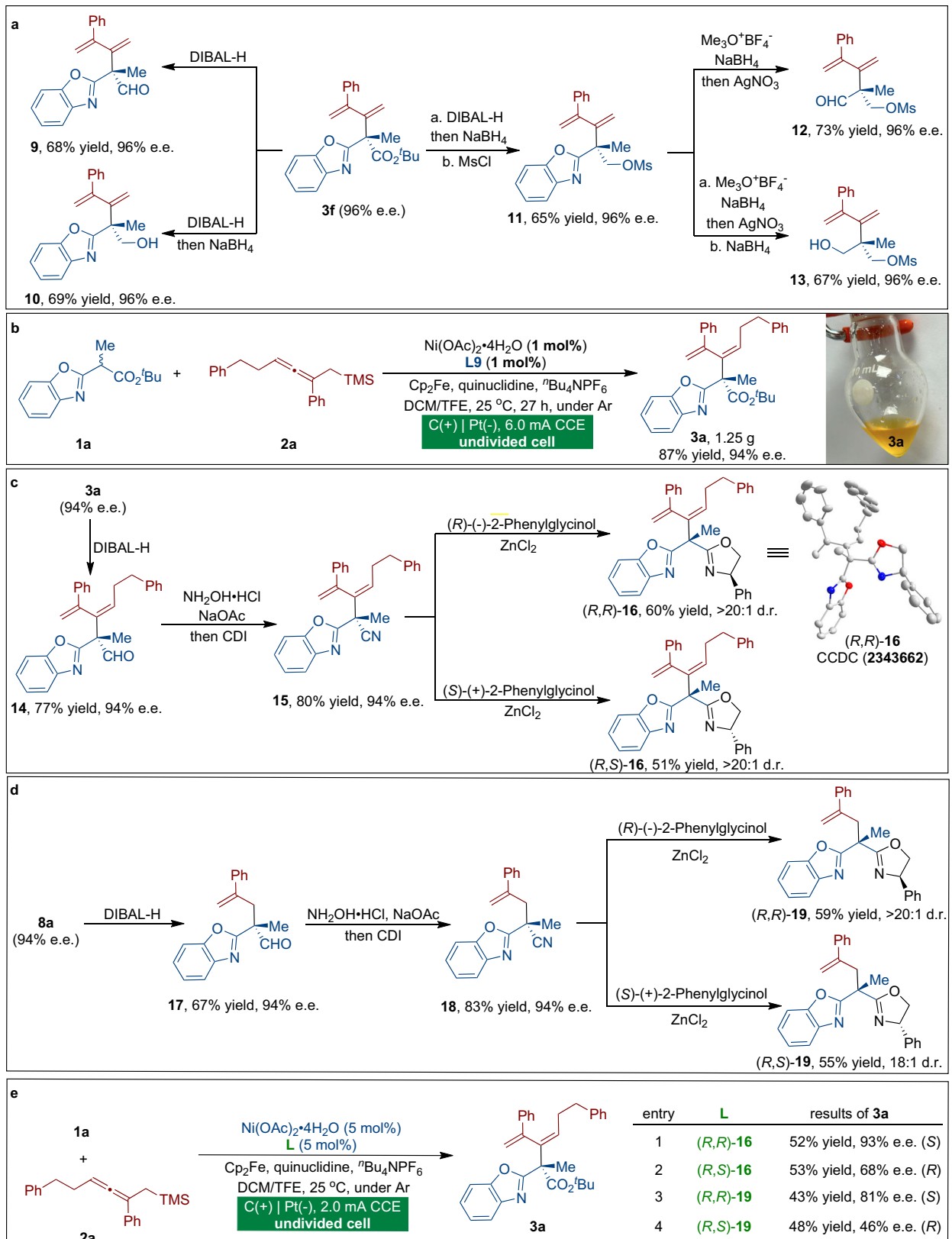

**Fig. 3 | Synthetic utility. a** Derivatization of dienylation product **3f. b** Large-scale reaction with a catalyst loading of 1 mol%. **c** Concise synthesis of chiral Boox ligands from dienylation product **3a. d** Concise synthesis of chiral Boox ligands from allylation product **8a. e** Evaluation of chiral Boox ligands in the enantioselective anodic oxidative dienylation reaction.

desired allylated product was obtained in 84% yield with 94% e.e. (**8a**). A larger scale reaction produced an analogous outcome (74% yield, 94% e.e.), highlighting the potential of the versatile nickel catalyst for asymmetric electrochemical synthesis. Screening different leaving groups (LG) of allylic reagents **7** provided no improvement in the yield or enantioselectivity of the reaction. With the optimized conditions identified, the substrate scope and generality of this reaction were then evaluated. The scope with respect to allylmethylsilanes **7** was examined. Aryl- and alkene-substituted allylmethylsilanes worked well without any losses in reaction efficiency or enantiocontrol (**8b**–**8m**). Moreover, the structure of **8l** was clearly verified through X-ray analysis, and its absolute configuration was designated as the (*S*)-configuration. Potassium allyltrifluoroborate exhibited good reactivity, resulting in the formation of the corresponding adduct in 63% yield with 87% enantioselectivity (**8n**). Furthermore, various benzoxazolyl acetates **1** performed well in this transformation, resulting in a range of allylated products with good enantioselectivities (**8o**–**8v**).However, the enantioselectivities exhibit a significant decrease when employing sterically substituted benzoxazolyl acetates (**8w** and **8x**). Further investigation revealed the compatibility of this method with various ester groups of benzoxazolyl acetates (**8y**–**8aa**).

To demonstrate the synthetic utility of the electrochemical radical coupling reactions, dienylated product **3f** can be easily modified in various ways (Fig. 3a). For instance, compound **3f** was transformed to the corresponding aldehyde **9** and alcohol **10** in good yields without any loss of enantioselectivity. The ester group of **3f** was smoothly converted to the corresponding oxymethyl sulfonyl moiety smoothly, delivering **11** in 65% yield and with 96% e.e. Furthermore, the benzoxazole moiety was easily removed to yield aldehyde **12** and alcohol **13** in high yields while maintaining the enantioselectivity, confirming the synthetic utility of the current protocol. As shown in Fig. 3b, the gram-scale synthesis of product **3a** was conducted on a 3 mmol scale under a very low catalyst loading (1 mol%), and a satisfactory isolated yield (87% yield, 1.25 g) was achieved with only slight erosion of enantioselectivity (94% e.e.).

The utility of the current methodology is further demonstrated in the generation of chiral benzoxazole-oxazoline (Boox) ligands for asymmetric catalysis. Tang[46] and Gade[47] have proven that introducing an additional group to the bridge carbon can enhance the stereocontrol capacities of chiral Box ligands in asymmetric catalysis. Using the anodic oxidative coupling strategy, we were able to synthesize Boox ligands in a uniform synthetic sequence, as shown in Figs. 3c and 3d. Initially, reduction of the ester group with DIBAL-H afforded aldehyde **14** in good yield. Aldehyde **14** was subsequently treated with hydroxylammonium chloride followed by heating at 80°C in the presence of carbonyldiimidazole to produce nitrile **15**, which was treated with two configurations of phenylglycinol to produce (*R,R*)-**16** and (*R,S*)-**16**, respectively (Fig. 3c). The above continuous process was also applied in the synthesis of (*R,R*)-**19** and (*R,S*)-**19**, and the desired products were obtained in good yields with high diastereocontrol (Fig. 3d). The obtained Boox **16** and **19** were then evaluated as chiral ligands in our anodic oxidative dienylation reactions (Fig. 3e). Surprisingly, product **3a** can be obtained in 52% yield and 93% e.e. by using (*R,R*)-**16** as a chiral ligand with a nickel catalyst (Fig. 3e, entry 1). Meanwhile, changes in absolute configurations or substituents on the bridging carbon have a significant impact on enantioselectivity control (entries 2-4 vs 1)[46,47]. The elaborate stereochemistry of Boox provides an opportunity to easily and efficiently manipulate the chiral space and electronic properties, making it a promising field for organometallics and asymmetric catalysis[48]. We believe that our methodology will serve as a paradigm for the asymmetric synthesis of diverse categories of chiral benzooxazole complexes and will offer more opportunities to serve as chiral ligands in asymmetric catalysis.

In summary, we have established the efficacy of electricity-driven asymmetric catalysts in suppressing racemic background reactions and precisely controlling the stereochemistry of oxidative dienylation and allylation processes. Our approach employs a chiral Lewis acid as a multifunctional catalyst, activating SET via electrochemical pathways, resulting in good yields, substrate compatibility, and enantioselectivities. Furthermore, unique chiral Boox ligands were created to demonstrate efficient performance in anodic oxidative dienylation, allowing access to catalysts capable of improving reaction stereoselectivity and efficiency. This innovative platform broadens the possibilities for designing robust enantioselective radical transformations, accelerating the creation of potent and stereocontrolled electrochemical reactions.

## Methods

### General procedure for the synthesis of products 3

In a dried sealed tube, $Ni(OAc)_2 \cdot 4H_2O$ (2.48 mg, 0.01 mmol) and **L9** (3.6 mg, 0.01 mmol) were dissolved in DCM (1.0 mL) under $N_2$ atmosphere, and the mixture was stirred for 1 h at room temperature before use. A 10 mL flask equipped with a magnetic stir bar was charged with **1** (0.1 mmol), **2** (0.3 mmol), $^nBu_4NPF_6$ (0.15 mmol), quinuclidine (0.05 mmol), $Cp_2Fe$ (0.01 mmol). The flask was equipped with a carbon rod (d = 6 mm) as the anode and a platinum plate (1.0 cm × 1.0 cm × 0.2 mm) as the cathode. The reaction mixture was degassed via vacuum evacuation and backfilled with argon three times, followed by the addition via a syringe of nickel catalyst solution made in advance, DCM (1.0 mL) and TFE (1.0 mL). The constant current (2.0 mA) electrolysis was carried out at 25 °C for 3 h. After completion, The solution was diluted with ethyl acetate and washed with saturated $NH_4Cl$ aqueous solution followed by distilled water washes. The aqueous phase was then extracted three times with ethyl acetate. The combined organic phase was dried over anhydrous $MgSO_4$ and concentrated under reduced pressure. The residue was purified by silica gel chromatography to afford the desired product **3**.

### General procedure for the synthesis of products 8

In a dried sealed tube, $Ni(OAc)_2 \cdot 4H_2O$ (2.5 mg, 0.01 mmol) and **L9** (3.6 mg, 0.01 mmol) were dissolved in DCM (1.0 mL) under $N_2$ atmosphere, and the mixture was stirred for 1 h at room temperature before use. A 10 mL flask equipped with a magnetic stir bar was charged with **1** (0.1 mmol), **7** (0.3 mmol), $^nBu_4NPF_6$ (0.15 mmol), quinuclidine (0.05 mmol), $Cp_2Fe$ (0.01 mmol). The flask was equipped with a carbon rod (d = 6 mm) as the anode and a platinum plate (1.0 cm × 1.0 cm × 0.2 mm) as the cathode. The reaction mixture was degassed via vacuum evacuation and backfilled with argon three times, followed by the addition via a syringe of nickel catalyst solution made in advance, DCM (0.5 mL), TFE (1.0 mL), and MeOH (0.5 mL). The constant current (2.0 mA) electrolysis was carried out at 25 °C for 3 h. After completion, The solution was diluted with ethyl acetate and washed with saturated $NH_4Cl$ aqueous solution, followed by distilled water washes. The aqueous phase was then extracted three times with ethyl acetate. The combined organic phase was dried over anhydrous $MgSO_4$ and concentrated under reduced pressure. The residue was purified by silica gel chromatography to afford the desired product **8**.

## Data availability

The X-ray crystallographic coordinates for the structures of **3f**, **8l**, and (*R,R*)-**16** have been deposited at the Cambridge Crystallographic Data Center (CCDC) under deposition nos. CCDC 2268498, CCDC 2268499 and CCDC 2343662, respectively. The data can be obtained free of charge from the Cambridge Crystallographic Data Centre via http://www.ccdc.cam.ac.uk/data_request/cif. The experimental procedures and characterization of all new compounds are provided in Supplementary Information. The authors declare that all other data supporting the findings of this study are available within this Article and its Supplementary Information. All data are available from the corresponding author upon request. Source data are provided with this

paper. NMR data in a mnova file format and HPLC traces are available at Zenodo at https://zenodo.org/records/10951810, under the Creative Commons Attribution 4.0 International license. Source data are provided with this paper.

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

## Acknowledgements

The authors acknowledge financial support from the National Key R&D Program of China (2023YFA1506700), National Natural Science Foundation of China (grant no. 22222113), CAS Project for Young Scientists in Basic Research (YSBR-054), and the Chinese Postdoctoral Science Foundation (2022TQ0324, 2023M733376, Q.Z.).

## Author contributions

C.G. conceived and designed the experiments. Q.Z. performed the experiments, analyzed the data, and prepared the Supplementary Information. J.Z., W.Z. and R.L. synthesized some of the substrates and ligands. C.G. wrote the paper. All authors discussed the results and commented on the manuscript.

## Competing interests

The authors declare no competing interests.
