## [Peer Review File · Nature Communications]

Enantioselective Nickel-Catalyzed Anodic Oxidative Dienylation and Allylation ReactionsReviewers' Comments:

Reviewer #1:

Remarks to the Author:

In this article, a nickel-based chiral catalyst was used as Lewis acid to promote SET of the carbonyl substrate at the anode, leading to generate an α -radical and react with both allylsilanes and allenylsilanes to form the corresponding allylation and dienylantion products. Ir- and Rh-based catalysts have been used previously for such Lewis acid-promoted electrochemical SET reactions at the carbonyl α position. The authors have advanced this field by using Ni-based catalysts. In their previous work, acyl imidazoles were used as substrates, whereas in this paper, α -oxazole-substituted esters were used as substrates, allowing significant expansion of the reaction scope. Meanwhile, the presence of the oxazole structure in the product also provided a possibility of late-stage modification to synthesize chiral oxazole ligands, a family of useful compounds.

The manuscript can be accepted after some minor revisions.

- (1) For substrate scope, how about other types of carbonyl groups, such as amides, ketones?
- (2) Second, for some cases with moderate yields, did the authors optimize the reaction with other reaction parameters. (In the condition table, the authors only showed the screening of chiral ligands)
- (3) In the mechanism study, strong background reaction was observed without a chiral Ni catalyst. Can the authors provide comments in the article to explain why the addition of a catalyst allows control of the chirality?
- (4) According to the experimental evidence given in the article, is there a possibility that the oxidation rate of the current (in terms of current magnitude) was far smaller than the Ni-catalyzed reaction rate, and could an experiment on the relationship between different current magnitudes and product ee values be added at a very small catalyst loading?
- (5) For the constant voltage reaction, is it possible that using a voltage greater than the substrate's intrinsic oxidation potential (over 1.98 V) would better emphasize the important role that the catalyst plays in the reaction?

Reviewer #2:

Remarks to the Author:

Chiral nickel catalyzed dienylation and allylation of benzoxazolyl acetates under anodic oxidative conditions are reported in this communication. The reaction shows good scope with respect to benzoxazolyl acetates, allenyl silanes and allyl silanes. The yields for the different transformations are modest at best. Through CV and other control studies, a radical mechanism is proposed for the dienylation.

The authors clearly demonstrate various functional group manipulations of the products including the synthesis of benzoxazolyl-oxazolines (Boox) ligands. Overall, this is a nice piece of enantioselective radical chemistry and the manuscript merits publication after appropriate modifications.

Comments/Questions:

Page 1 Ln 12-14: Rephrase the sentence

Figure 1a: tunable ligands; also spelling for quaternary stereocenter

Page 2 Ln 75: use "on a range of" instead of "of a range of".

Page 3: Figure 1b: if the authors can indicate the suppression of racemic reaction by color coding that part of the scheme in gray color it would be helpful.

Page 3 Ln 81-85: Figure caption: (a) should also indicate anionic pathway; (b) ; (c) don't understand what is divergent about this methodology. It is just that somophiles are different; (d) were other bases tried in place of quinuclidine? Is an aliphatic base required?

Page 3 Figure 1d: Is there any E/Z ratio or the double bond geometry is as indicated in the product? If there is E/Z ratio, indicate them in the results.

Page 5 Ln 115: The authors should explain how the experiments with chemical oxidants without electrical input was done (in figure 2 or in ESI and refer to those conditions in the text).

Page 5 Ln 131: use inhibited instead of repressed.

Page 4 Figure 2h: Indicate how E/Z ratios were determined.

Page 6 Ln 156: use "excellent" instead of "superb".

Page 7 Table 1: Is there any E/Z ratio or the double bond geometry is as indicated in the product? Indicate how double bond geometry was determined. If there is E/Z ratio indicate them for each product.

For Table 1: Authors should include effect of sterics around α -carbon: (isopropyl, cyclohexyl and tert-butyl substitutions). The authors should also include results of aryl and heteroaryl substituents on α -carbon.

It is interesting that the size of the ester makes a difference in the product ee. A comment in the text would be appropriate.

Page 8 Table 2: products 8b-8e: what is X? Is it R instead of X?

For Table 2: Authors should include effect of sterics around α -carbon: (isopropyl, cyclohexyl and tert-butyl substitutions). The authors should also include results of aryl and heteroaryl substituents on α -carbon.

The authors have reported the leaving group effect for allylation reaction, but not for reactions with allenes. Why?

Benzoxazolyl acetates are reported in the study. Do other substituents such as amides, ketones, imides work?

Does benzimidazole or benzothiazole auxiliary work? How about a simple oxazole in place of benzoxazole?

Reviewer #3:

Remarks to the Author:

Minor revisions as follows:

1. The abstract and the introduction give the impression that there is little precedent in enantioselective Lewis acid catalysis with radical intermediates. In fact, this is an established strategy. The introduction should mention the key reports in this area so that the reader can properly contextualize the study.

For example:

Huang, X.; Zhang, Q.; Lin, J.; Harms, K.; Meggers, E., Electricity-driven asymmetric Lewis acid catalysis. *Nat. Catal.* 2019, 2 (1), 34-40.,

Zhang, Q.; Liang, K.; Guo, C., Enantioselective Nickel-Catalyzed Electrochemical Radical Allylation. *Angew. Chem. Int. Ed.* 2022, 61 (38), e202210632.

2. In the abstract: The term "ingenious" may be somewhat overly enthusiastic. A more tempered expression should be chosen.

3. In line 50: "Incredibly low potential" is not a scientific expression. This could be rephrased.

4. In line 62: Typo: "anode oxidation conditions"

5. In line 63: "Exceptional yields". In conclusion: "remarkable yields"

These are not suitable descriptions. The yields are more in the range of 50-60%, and yields over 90% are not achieved.

6. Caption of Fig. 1: "D" should be "d"

7. The gram-scale reaction should be discussed more in the context of synthetic utility rather than within the mechanistic studies.

8. Neither for the substrate scope nor the gram-scale reaction times are provided. However, this is a crucial reaction parameter for reproducing the reaction and determining the energy efficiency (Faradaic yield).

9. In the catalytic cycle, no specific information is provided regarding the fate of the TMS group. What do the authors suggest? Is it released as a TMS cation or as a TMS radical? Is TMA-OCH₂CF₃ obtained as a stoichiometric byproduct? If the TMS radical is released, anodic oxidation would not be

immediately necessary. The authors should provide an explanation here.

10. To determine the role of ferrocene, it would be interesting to test whether Cp₂Fe^{III}(PF₆) is effective as a stoichiometric oxidizing agent for the reaction.

11. The role of quinuclidine is not explicitly explained. Is it possible that it serves as a HAT (Hydrogen Atom Transfer) mediator and is involved in the H-abstraction? The authors should clarify their position on this.

12. In Fig. 2j: ½ H₂.

13. In line 156: "Superb" is a too positive term for the actual yields obtained.

14. Fig 3b and 3c: Typo "Plenylglycinol"

15. What are the d.r. values for the synthesized Boox ligands? Since the nitriles (compound 15 and 18) were not used with >99% ee, diastereomers should be obtained. Could these be separated chromatographically? If so, the Boox ligands would likely be more easily accessible even from the racemic nitriles.

16. In conclusion: "outstanding performance" is too positive. The yields and ee values are not that high.

Supporting Information

1. Page S12 and S13: No reaction times for the general procedures are provided

2. Page S26: No reaction times for the gram scale is provided

3. Page S30 and S31: Typo "Plenylglycinol", "Plenylglycinolin"

4. Page S47 and S48: The Flack parameters are not mentioned, although they are crucial for determining the absolute configurations.

Responses to Comments

for

Enantioselective Nickel-Catalyzed Anodic Oxidative Dienylation and Allylation Reactions

Qinglin Zhang, Jiayin Zhang, Wangjie Zhu, Ruimin Lu, Chang Guo *

Please find below a list of comments and changes made to the above manuscript in response to reviewers. Further changes made to the manuscript since submission are also listed at the end of this document. **A copy of the revised manuscript, a word document showing tracked changes** made to the manuscript since submission, and **revised supplementary information** are also included as part of this revision.

Reply to comments by Reviewer 1

We appreciate Reviewer 1 for the favorable comments and many helpful suggestions!

(1) In this article, a nickel-based chiral catalyst was used as Lewis acid to promote SET of the carbonyl substrate at the anode, leading to generate an α -radical and react with both allylsilanes and allenylsilanes to form the corresponding allylation and dienylation products. Ir- and Rh-based catalysts have been used previously for such Lewis acid-promoted electrochemical SET reactions at the carbonyl α position. The authors have advanced this field by using Ni-based catalysts. In their previous work, acyl imidazoles were used as substrates, whereas in this paper, α -oxazole-substituted esters were used as substrates, allowing significant expansion of the reaction scope. Meanwhile, the presence of the oxazole structure in the product also provided a possibility of late-stage modification to synthesize chiral oxazole ligands, a family of useful compounds. The manuscript can be accepted after some minor revisions.

Answer: We appreciate Reviewer 1 for the favorable comments and helpful suggestions! These comments are greatly valuable and helpful for revising and improving our paper. We have made all the necessary amendments as suggested in our revised manuscript and revised supplementary information.

(2) For substrate scope, how about other types of carbonyl groups, such as amides, ketones?

Answer: As suggested by Reviewer 1, we have tried different kinds of carbonyl groups, including amides (**S1a**, **S1b**, **S1d**), imides **S1c**, and ketone **S1e** for the reaction. As shown in Table S2 (Page S13 in our revised supplementary information), the use of different amine partners showed dramatic effect on the reactivity (entries 1-4). No reaction occurs in the presence of amides **S1a**, **S1b**, **S1c** (entries 1-3). Amide **S1d** proved to be suitable partner, delivering the desired product **3aa** in 41% yield and 94% e.e. (entry 8). Ketone **S1e** exhibited good reactivity, resulting in the formation of the corresponding adduct **3z** in 59% yield with 78% enantioselectivity (entry 9). We

have included these results in our revised manuscript (**3z** and **3aa**) and supplementary information (Page S13).

Table S2: Survey of substituent of benzoxazolyl acetates ^a

S1a

S1b

S1c

S1d

S1e

Entry	Substrates S1	3	Yield (%)	e.e. (%) of 3
1	S1a	—	nr	—
2	S1b	—	nr	—
3	S1c	—	nr	—
4	S1d	3aa	36	92
5 ^b	S1d	3aa	42	92
6 ^c	S1d	3aa	37	92
7 ^d	S1d	3aa	nr	—
8 ^e	S1d	3aa	41	94
9 ^e	S1e	3z	59	78

^aReactions were carried out with **S1** (0.1 mmol), **2a** (0.3 mmol), ^b*t*Bu₄NPF₆ (0.15 mmol), quinuclidine (0.05 mmol), Ni(OAc)₂·4H₂O (10 mol%), **L9** (10 mol%), Cp₂Fe (10 mol%) and DCM/TFE = 2:1 (3 mL) at 25 °C in an undivided cell. ^b**2a** (0.6 mmol), 50 °C. ^c**2a** (0.6 mmol), HCOOH (20 mol%), ^dNi(OAc)₂·4H₂O (20 mol%), **L9** (20 mol%), **2a** (0.6 mmol), 50 °C. TFE, 2,2,2-trifluoroethanol. nr = no reaction.

(*S,E*)-2-(benzo[d]oxazol-2-yl)-*N*,2-dimethyl-6-phenyl-3-(1-phenylvinyl)hex-3-enamide (3aa**)**

Reaction time: 3 h. ¹H NMR (400 MHz, CDCl₃) δ 8.39 (q, *J* = 4.1 Hz, 1H), 7.62 – 7.52 (m, 1H), 7.42 – 7.35 (m, 1H), 7.32 – 7.26 (m, 3H), 7.25 – 7.20 (m, 2H), 7.19 – 7.14 (m, 1H), 7.14 – 7.08 (m, 2H), 7.06 – 7.00 (m, 4H), 5.94 (t, *J* = 7.2 Hz, 1H), 5.41 (d, *J* = 1.4 Hz, 1H), 4.68 (d, *J* = 1.5 Hz, 1H), 2.82 – 2.65 (m, 5H), 2.49 – 2.37 (m, 2H), 1.82 (s, 3H). ¹³C NMR (100 MHz, CDCl₃) δ 170.26, 166.98, 149.98, 145.09, 141.62, 140.62, 140.04, 138.92, 131.25, 128.82, 128.38, 127.96, 127.38, 126.10, 126.01, 125.11, 124.39, 119.78, 117.11, 110.66, 54.29, 35.93, 31.93, 26.66, 22.23. **ESI-MS**: calculated [C₂₉H₂₈N₂O₂ + H]⁺: 437.2224, found: 437.2228. [α]_D²⁰ = -20.80 (c = 0.52, CH₂Cl₂). The product was analyzed by HPLC to determine the enantiomeric excess: 94% e.e. (CHIRALPAK IE, hexane/*i*-PrOH = 80/20, detector: 254 nm, T = 25 °C, flow rate: 1 mL/min), t₁(major) = 9.01 min, t₂(minor) = 10.23 min.

(*S,E*)-2-(benzo[d]oxazol-2-yl)-2-methyl-1,6-diphenyl-3-(1-phenylvinyl)hex-3-en-1-one (3z**)**

Reaction time: 3 h. ¹H NMR (400 MHz, CDCl₃) δ 7.88 – 7.83 (m, 2H), 7.62 – 7.58 (m, 1H), 7.51 – 7.45 (m, 2H), 7.40 – 7.34 (m, 2H), 7.34 – 7.27 (m, 2H), 7.21 – 7.13 (m, 3H), 7.11 – 7.07 (m, 2H), 7.07 – 7.03 (m, 3H), 7.02 – 6.97 (m, 2H), 6.06 (t, *J* = 7.2 Hz, 1H), 5.59 (d, *J* = 1.3 Hz, 1H), 5.03 (d, *J* = 1.1 Hz, 1H), 2.77 – 2.66 (m, 2H), 2.62 – 2.50 (m, 2H), 1.82 (s, 3H). ¹³C NMR (100 MHz, CDCl₃) δ 198.60, 166.78, 150.91, 145.35, 141.54, 141.08, 139.54, 138.28, 135.93, 135.56, 132.56, 130.06, 128.70, 128.42, 128.24, 127.72, 126.40, 126.02, 124.88, 124.16, 120.15, 117.89, 110.76, 59.59, 35.80, 32.52, 22.99. **ESI-MS**: calculated [C₃₄H₂₉NO₂ + H]⁺: 484.2271, found: 484.2285. [α]_D²⁰ = +33.40 (c = 0.59, CH₂Cl₂). The product was analyzed by HPLC to determine the enantiomeric excess: 78% e.e. (CHIRALPAK IC,

hexane/*i*-PrOH =80/20, detector: 254 nm, T = 25 °C, flow rate: 1 mL/min), t_1 (major) = 5.00 min, t_2 (minor) = 9.97 min.

(3) Second, for some cases with moderate yields, did the authors optimize the reaction with other reaction parameters. (In the condition table, the authors only showed the screening of chiral ligands)

Answer: As suggested by Reviewer 1, we continued to optimize the reaction conditions for reactions with moderate yields. We found that adding formic acid could effectively increase the reaction yield and avoid the occurrence of dimerization side reactions. We have included these results in our revised manuscript.

Unless otherwise specified, all reactions were carried out using racemic benzoxazolyl acetate **1** (0.1 mmol), allenylmethylsilane **2** or allylic counterpart **7** (0.30 mmol), ⁿBu₄NPF₆ (0.15 mmol), quinuclidine (0.05 mmol), Ni(OAc)₂·4H₂O (10 mol%), **L9** (10 mol%), Cp₂Fe (10 mol%), and DCM/TFE = 2:1 (3 mL) at 25 °C for 3 h under constant-current conditions in an undivided cell. ^aHCO₂H (20 mol%). ^b**L4** (10 mol%). ^c6 h. ^dAllenylmethylsilane **2** (0.60 mmol).

In addition, we provided the results of optimizing reaction conditions, including solvent, base, temperature, electrolyte, and anode in our revised supplementary information (Page S16-S18, Table S9-S13).

Table S9: Survey of solvents^a

Entry	Solvent (3 mL)	Yield (%)	e.e. (%) of 3a
1	DCM	16	53
2	DCM/MeOH = 1:1	26	68
3	THF/MeOH = 1:1	13	39
4	MeCN/MeOH = 1:1	8	36
5	DCM/TFE = 1:1	65	97
6	DCM/TFE = 2:1	79	96

^aReactions were carried out with **1a** (0.1 mmol), **2a** (0.3 mmol), ^tBu₄NPF₆ (0.15 mmol), quinuclidine (0.05 mmol), Ni(OAc)₂·4H₂O (10 mol%), **L9** (10 mol%) and Cp₂Fe (10 mol%) at 25 °C in an undivided cell. TFE, 2,2,2-trifluoroethanol.

Table S10: Survey of base^a

Entry	Base (0.05 mmol)	Yield (%)	e.e. (%) of 3a
1	No base	51	96
2	2,6-dimethoxyppyridine	49	96
3	Et ₃ N	74	96
4	DMAP	57	95
5	quinuclidine	79	96
6	K ₂ CO ₃	73	95

^aReactions were carried out with **1a** (0.1 mmol), **2a** (0.3 mmol), ^tBu₄NPF₆ (0.15 mmol), base (0.05 mmol), Ni(OAc)₂·4H₂O (10 mol%), **L9** (10 mol%), Cp₂Fe (10 mol%) and DCM/TFE = 2:1 (3 mL) at 25 °C in an undivided cell. TFE, 2,2,2-trifluoroethanol.

Table S11: Survey of temperature^a

Entry	Temperature (°C)	Yield (%)	e.e. (%) of 3a
1	25	79	96
2	35	76	96
3	10	53	96
4	0	27	97
5	-10	22	97

^aReactions were carried out with **1a** (0.1 mmol), **2a** (0.3 mmol), ^tBu₄NPF₆ (0.15 mmol), quinuclidine (0.05 mmol), Ni(OAc)₂·4H₂O (10 mol%), **L9** (10 mol%), Cp₂Fe (10 mol%) and DCM/TFE = 2:1 (3 mL) in an undivided cell. TFE, 2,2,2-trifluoroethanol.

Table S12: Survey of electrolyte^a

Entry	Electrolyte (0.15 mmol)	Yield (%)	e.e. (%) of 3a
1	ⁿ Bu ₄ NPF ₆	79	96
2	ⁿ Bu ₄ NBF ₆	63	97
3	ⁿ Bu ₄ NBr	69	96
4	ⁿ Bu ₄ NClO ₄	63	96
5	ⁿ Bu ₄ NOAc	30	92

^aReactions were carried out with **1a** (0.1 mmol), **2a** (0.3 mmol), electrolyte (0.15 mmol), quinuclidine (0.05 mmol), Ni(OAc)₂·4H₂O (10 mol%), **L9** (10 mol%), Cp₂Fe (10 mol%) and DCM/TFE = 2:1 (3 mL) at 25 °C in an undivided cell. TFE, 2,2,2-trifluoroethanol.

Table S13: Survey of anode^a

Entry	anode	Yield (%)	e.e. (%) of 3a
1	Pt	67	96
2	C	79	96
3	BDD	65	96
4	RVC	55	97
5	Ni form	56	96

^aReactions were carried out with **1a** (0.1 mmol), **2a** (0.3 mmol), ⁿBu₄NPF₆ (0.15 mmol), quinuclidine (0.05 mmol), Ni(OAc)₂·4H₂O (10 mol%), **L9** (10 mol%), Cp₂Fe (10 mol%) and DCM/TFE = 2:1 (3 mL) at 25 °C in an undivided cell. TFE, 2,2,2-trifluoroethanol.

(4) In the mechanism study, strong background reaction was observed without a chiral Ni catalyst. Can the authors provide comments in the article to explain why the addition of a catalyst allows control of the chirality?

Answer: Great appreciation to the reviewer 1 for posing this significant comment. We attribute the effective suppression of background reactions upon the introduction of chiral catalysts primarily to two factors. Firstly, a significant reduction in the onset oxidation potential of **1a** was observed with the nickel complex and quinuclidine (+0.36 V versus SCE, Fig. 2b), indicating an easier formation of a catalyst-bound reactive intermediate to suppress racemic background reactions. Secondly, our anodic oxidative coupling methodology offers distinct advantages by enabling the tuning of oxidation potential and current density to achieve controllable oxidation processes, which proceed at a rate slower than that of the nickel-catalyzed reaction. Catalyst loadings varying from 0.5 mol% to 10 mol% produced coupling adduct **3a** with a similar enantiomeric excess in each case (Fig. 2f), indicating that the racemic background reaction contributes minimally to the observed stereoselectivity. In contrast, the utilization of silver oxide as a chemical oxidant instead of an electrical input revealed a noticeable decline in the enantiomeric excess of the coupling adduct **3a** as the catalyst loading decreased from 10 mol% to

0.5 mol%. (see revised supplementary information Page S53, Table S21). Through comparative experiments, it is evident that with the addition of stoichiometric silver oxide as the oxidant, the enantioselectivity of the reaction significantly decreases as the catalyst loading decreases. Furthermore, under electrochemical conditions, increasing the reaction current to enhance the rate of oxidation event while reducing the amount of catalyst leads to a partially decrease in the enantioselectivity of the reaction (see revised supplementary information Page S18, Table S14).

We have provided comments in the revised manuscript “Compared to conventional processes that rely on stoichiometric chemical oxidants, our anodic oxidative coupling methodology provides distinct advantages through its ability to tune oxidation potential and current density to achieve controllable oxidation processes, which are slower than the rate of the nickel-catalyzed reaction. Furthermore, the catalytically generated intermediates are inherently more susceptible to oxidation than the reactants (Fig. 2b, green line), also playing a crucial role in the suppression of racemic background reactions.”.

(5) According to the experimental evidence given in the article, is there a possibility that the oxidation rate of the current (in terms of current magnitude) was far smaller than the Ni-catalyzed reaction rate, and could an experiment on the relationship between different current magnitudes and product ee values be added at a very small catalyst loading?

Answer: As suggested by reviewer 1, we investigated the relationship between different current magnitudes and product e.e. values with a 5 mol% catalyst loading. We found that the enantioselectivity of the product **3a** gradually decreases with increasing current (Table S14). Under the reaction conditions of constant current at 10 mA, product **3a** was obtained with only 16% yield and 88% e.e. (entry 5). Therefore, increasing the reaction current to enhance the rate of oxidation event while reducing the amount of catalyst leads to a partially decrease in the enantioselectivity of the reaction. Particularly, a significant decrease in yield is observed when the current is markedly increased, attributed to the oxidative decomposition of the allenylmethylsilane substrate **2a** under these reaction conditions. The extra allenylmethylsilane **2a** disappears at 10 mA CCE, but when 2.0 mA CCE is employed, a portion of the **2a** still exists even after the reaction is completed. We have included these results in our revised supplementary information (Page S18, Table S14)

Table S14: Survey of current^a

Entry	CCE	Yield (%)	e.e. (%) of 3a
1	2.0 mA	76	96
2	4.0 mA	74	93
3	6.0 mA	51	92
4	8.0 mA	19	90
5	10.0 mA	16	88

^aReactions were carried out with **1a** (0.1 mmol), **2a** (0.3 mmol), ⁿBu₄NPF₆ (0.15 mmol), quinuclidine (0.05 mmol), Ni(OAc)₂·4H₂O (5 mol%), **L9** (5 mol%), Cp₂Fe (10 mol%) and DCM/TFE = 2:1 (3 mL) at 25 °C under constant-current conditions in an undivided cell. TFE, 2,2,2-trifluoroethanol.

(6) For the constant voltage reaction, is it possible that using a voltage greater than the substrate's intrinsic oxidation potential (over 1.98 V) would better emphasize the important role that the catalyst plays in the reaction?

Answer: Under a catalyst loading of 5 mol%, reactions were conducted using constant potentials of 0.4 V, 1.0 V, and 2.5 V (versus SCE) respectively. When the potential increases to 2.5 V (Table S15, entry 3 in revised supplementary information, Page S18), there is a partial decrease in the enantioselectivity of the product. Therefore, increasing the voltage for the reaction to enhance the rate of oxidation event while reducing the amount of catalyst leads to a partially decrease in the enantioselectivity of the reaction. Since the oxidation potential of allenylmethylsilane **2a** is significantly lower than that of substrate **1a**, **2a** is preferentially oxidized as the potential increases, thus avoiding the oxidation process of **1a** which might induce the racemic reaction and resulting in a significant decrease in yield.

Table S15: Survey of voltage^a

Entry	CPE (vs SCE)	Yield (%)	e.e. (%) of 3a
1	0.4 V	67	96
2	1.0 V	72	95
3	2.5 V	12	92

^aReactions were carried out with **1a** (0.1 mmol), **2a** (0.3 mmol), ⁿBu₄NPF₆ (0.15 mmol), quinuclidine (0.05 mmol), Ni(OAc)₂·4H₂O (5 mol%), **L9** (5 mol%), Cp₂Fe (10 mol%) and DCM/TFE = 2:1 (3 mL) at 25 °C under constant-voltage conditions in an undivided cell. TFE, 2,2,2-trifluoroethanol. CPE, constant potential electrolysis.

The onset potential of [**1a** + quinuclidine + Ni/(*R,R*)-**L9**] was measured approximately at +0.36 V (versus SCE). The onset potential of **1a** was measured approximately at +1.98 V (versus SCE). The onset potential of **2a** was measured approximately at +1.11 V (versus SCE). We have included these results in our revised supplementary information (Page S44-S46).

Reply to comments by Reviewer 2

We appreciate Reviewer 2 for the favorable comments and helpful suggestions!

- (1) Chiral nickel catalyzed dienylation and allylation of benzoxazolyl acetates under anodic oxidative conditions are reported in this communication. The reaction shows good scope with respect to benzoxazolyl acetates, allenyl silanes and allyl silanes. The yields for the different transformations are modest at best. Through CV and other control studies, a radical mechanism is proposed for the dienylation. The authors clearly demonstrate various functional group manipulations of the products including the synthesis of benzoxazolyl-oxazolines (Boox) ligands. Overall, this is a nice piece of enantioselective radical chemistry and the manuscript merits publication after appropriate modifications.

Answer: We appreciate Reviewer 2 for the favorable comments and helpful suggestions! Those comments are greatly valuable and helpful for revising and improving our paper. We have made all the necessary amendments as suggested in our revised manuscript and revised supplementary information.

- (2) Page 1 Ln 12-14: Rephrase the sentence

Answer: As suggested by reviewer 2, We have rephrased the sentence “This methodology facilitates both asymmetric dienylation and allylation reactions, resulting in the formation of all-carbon quaternary stereocenters and demonstrating significant potential in the modular synthesis of functional and chiral benzoxazole-oxazoline (Boox) ligands.” in our revised manuscript.

- (3) Figure 1a: tunable ligands; also spelling for quaternary stereocenter

Answer: As suggested by reviewer 2, We have corrected these errors in our revised manuscript: “Readily tunable of chiral ligands” was changed to “Readily tunable chiral ligands”; “quaternary stereocenter” was changed to “quaternary stereocenter”.

- (4) Page 2 Ln 75: use “on a range of” instead of “of a range of”.

Answer: As suggested by reviewer 2, We have changed “of a range of” to “on a range of” in our revised manuscript.

- (5) Page 3: Figure 1b: if the authors can indicate the suppression of racemic reaction by color coding that part of the scheme in gray color it would be helpful.

Answer: As suggested by reviewer 2, We have adjusted the color of the suppression of racemic reaction to gray in our revised manuscript.

- (6) Page 3 Ln 81-85: Figure caption: (a) should also indicate anionic pathway; (b) ; (c) don't understand what is divergent about this methodology. It is just that somophiles are different;

Answer: As suggested by reviewer 2, we have added “anionic pathway” in the figure caption of

Figure 1a; Additionally, we have removed “divergent” in the figure caption of Figure 1c in our revised manuscript.

(7) were other bases tried in place of quinuclidine? Is an aliphatic base required?

Answer: As suggested by reviewer 2, we investigated the use of different types of bases in the reaction (Table S10). The reaction proceeded even without the addition of a base, affording **3a** in 51% and 96% e.e. (entry 1). Other organic bases such as 2,6-dimethylpyridine, triethylamine, or DMAP also yielded reaction outcomes similar to quinuclidine (entries 2-4 vs 5). Inorganic bases also effectively promoted the reaction, resulting **3a** in 73% and 95% e.e. (entry 6). These results indicate that various types of bases can effectively promote the reaction. We have included these results in our revised supplementary information (Page S17, Table S10).

Table S10: Survey of base^a

Entry	Base	Yield (%)	e.e. (%) of 3a
1	No base	51	96
2	2,6-dimethoxyypyridine	49	96
3	Et ₃ N	74	96
4	DMAP	57	95
5	quinuclidine	79	96
6	K ₂ CO ₃	73	95

^aReactions were carried out with **1a** (0.1 mmol), **2a** (0.3 mmol), ^tBu₄NPF₆ (0.15 mmol), base (0.05 mmol), Ni(OAc)₂·4H₂O (10 mol%), **L9** (10 mol%), Cp₂Fe (10 mol%) and DCM/TFE = 2:1 (3 mL) at 25 °C in an undivided cell. TFE, 2,2,2-trifluoroethanol.

(8) Page 3 Figure 1d: Is there any E/Z ratio or the double bond geometry is as indicated in the product? If there is E/Z ratio, indicate them in the results.

Answer: By comparing the crude NMR spectra as well as the spectra from the purified compound, we only detected the product **3a** with a single configuration.

¹H NMR spectrum of crude **3a**

¹H NMR spectrum of **3a**

Comparison of ^1H NMR spectrum of **3a** (blue) and crude **3a** (red)

The NOESY spectrum of **3a** suggests that product **3a** is in the *E* configuration.

NOESY spectrum of **3a**

In order to further confirm the configuration of **3a**, we obtained the single crystal structure of product (*R,R*)-**16** (CCDC 2343662), providing additional evidence that product **3a** is in the *E* configuration. We have included these results in our revised manuscript and supplementary information.

(9) Page 5 Ln 115: The authors should explain how the experiments with chemical oxidants without electrical input was done (in figure 2 or in ESI and refer to those conditions in the text).

Answer: We appreciate Reviewer 2 for the helpful suggestions. We have included “The reaction condition with silver oxide as a chemical oxidant instead of an electrical input: reactions were carried out using racemic benzoxazolyl acetate **1a** (0.1 mmol), allenylmethylsilane **2a** (0.30 mmol), $^t\text{Bu}_4\text{NPF}_6$ (0.15 mmol), quinuclidine (0.05 mmol), $\text{Ni}(\text{OAc})_2\cdot 4\text{H}_2\text{O}/\text{L9}$ (0.5-10 mol%), Cp_2Fe (0.5-10 mol%), Ag_2O (0.30 mmol), and $\text{DCM}/\text{TFE} = 2:1$ (3 mL) at 25 °C.” in our revised manuscript. In addition, we included the reaction details in the revised supplementary information (Page S51-S53, Table S21). As shown below:

Investigation of the catalyst loading with Ag_2O instead of electric current under the standard condition

In a dried sealed tube, $\text{Ni}(\text{OAc})_2\cdot 4\text{H}_2\text{O}$ (4.96 mg, 0.02 mmol) and **L9** (7.2 mg, 0.02 mmol) were dissolved in DCM (2.0 mL) under N_2 atmosphere, and the mixture was stirred for 1 h at room temperature before use.

Ni/(*R,R*)-L9 (0.5 mol%): A 10 mL flask equipped with a magnetic stir bar was charged with **1a** (0.1 mmol), **2a** (0.3 mmol), $^t\text{Bu}_4\text{NPF}_6$ (0.15 mmol), quinuclidine (0.05 mmol), Cp_2Fe (0.0005 mmol) and Ag_2O (0.3 mmol). The reaction mixture was degassed via vacuum evacuation and backfilled with argon three times, followed by the addition via a syringe of nickel catalyst solution made in advance (50 μL), DCM (2.0 mL), and TFE (1.0 mL). The mixture was stirred at 25 °C until complete consumption of the substrate (monitored by TLC). The solution was diluted with ethyl acetate and washed with saturated NH_4Cl aqueous solution followed by distilled water washes. The aqueous phase was then extracted three times with ethyl acetate. The combined organic phase was dried over anhydrous MgSO_4 and concentrated under reduced pressure. The residue was purified by silica gel chromatography to afford the desired product **3a**.

Ni/(*R,R*)-L9 (1 mol%): A 10 mL flask equipped with a magnetic stir bar was charged with **1a** (0.1 mmol), **2a** (0.3 mmol), $^t\text{Bu}_4\text{NPF}_6$ (0.15 mmol), quinuclidine (0.05 mmol), Cp_2Fe (0.001 mmol) and Ag_2O (0.3 mmol). The reaction mixture was degassed via vacuum evacuation and backfilled with argon three times, followed by the addition via a syringe of nickel catalyst solution made in advance (100 μL),

DCM (2.0 mL), and TFE (1.0 mL). The mixture was stirred at 25 °C until complete consumption of the substrate (monitored by TLC). The solution was diluted with ethyl acetate and washed with saturated NH₄Cl aqueous solution followed by distilled water washes. The aqueous phase was then extracted three times with ethyl acetate. The combined organic phase was dried over anhydrous MgSO₄ and concentrated under reduced pressure. The residue was purified by silica gel chromatography to afford the desired product **3a**.

Ni/(*R,R*)-L9 (2 mol%): A 10 mL flask equipped with a magnetic stir bar was charged with **1a** (0.1 mmol), **2a** (0.3 mmol), ⁿBu₄NPF₆ (0.15 mmol), quinuclidine (0.05 mmol), Cp₂Fe (0.002 mmol) and Ag₂O (0.3 mmol). The reaction mixture was degassed via vacuum evacuation and backfilled with argon three times, followed by the addition via a syringe of nickel catalyst solution made in advance (200 uL), DCM (2.0 mL), and TFE (1.0 mL). The mixture was stirred at 25 °C until complete consumption of the substrate (monitored by TLC). The solution was diluted with ethyl acetate and washed with saturated NH₄Cl aqueous solution followed by distilled water washes. The aqueous phase was then extracted three times with ethyl acetate. The combined organic phase was dried over anhydrous MgSO₄ and concentrated under reduced pressure. The residue was purified by silica gel chromatography to afford the desired product **3a**.

Ni/(*R,R*)-L9 (5 mol%): A 10 mL flask equipped with a magnetic stir bar was charged with **1a** (0.1 mmol), **2a** (0.3 mmol), ⁿBu₄NPF₆ (0.15 mmol), quinuclidine (0.05 mmol), Cp₂Fe (0.005 mmol) and Ag₂O (0.3 mmol). The reaction mixture was degassed via vacuum evacuation and backfilled with argon three times, followed by the addition via a syringe of nickel catalyst solution made in advance (0.5 mL), DCM (1.5 mL), and TFE (1.0 mL). The mixture was stirred at 25 °C until complete consumption of the substrate (monitored by TLC). The solution was diluted with ethyl acetate and washed with saturated NH₄Cl aqueous solution followed by distilled water washes. The aqueous phase was then extracted three times with ethyl acetate. The combined organic phase was dried over anhydrous MgSO₄ and concentrated under reduced pressure. The residue was purified by silica gel chromatography to afford the desired product **3a**.

Ni/(*R,R*)-L9 (10 mol%): A 10 mL flask equipped with a magnetic stir bar was charged with **1a** (0.1 mmol), **2a** (0.3 mmol), ⁿBu₄NPF₆ (0.15 mmol), quinuclidine (0.05 mmol), Cp₂Fe (0.01 mmol) and Ag₂O (0.3 mmol). The reaction mixture was degassed via vacuum evacuation and backfilled with argon three times, followed by the addition via a syringe of nickel catalyst solution made in advance (1.0 mL), DCM (1.0 mL), and TFE (1.0 mL). The mixture was stirred at 25 °C until complete consumption of the substrate (monitored by TLC). The solution was diluted with ethyl acetate and washed with saturated NH₄Cl aqueous solution followed by distilled water washes. The aqueous phase was then extracted three times with ethyl acetate. The combined organic phase was dried over anhydrous MgSO₄ and concentrated under reduced pressure. The residue was purified by silica gel chromatography to afford the desired product **3a**.

Table S21: Investigation of the catalyst loading with Ag₂O instead of electric current under the standard condition

Entry	Ni/(R,R)-L9 (mol %)	Yield (%)	e.e. (%) of 3a
1	0.5	30	11
2	1	38	32
3	2	41	49
4	5	52	79
5	10	60	87

(10) Page 5 Ln 131: use inhibited instead of repressed.

Answer: As suggested by reviewer 2, We have changed “repressed” to “inhibited” in our revised manuscript.

(11) Page 4 Figure 2h: Indicate how *E/Z* ratios were determined.

Answer: The main products of compounds **5** and **6** were separately analyzed by NOESY spectra, inferring that the major structures of compounds **5** and **6** are in the *E* configuration. Through crude spectral analysis, it was determined that the *E/Z* ratios of compound **5** and **6** were 3.8:1 and 2.4:1, respectively.

¹H NMR spectrum of **5** (*E/Z* = 3.8:1)

¹H NMR spectrum of **5** (major isomer)

NOESY spectrum of **5** (major isomer)

¹H NMR spectrum of **6** (E/Z = 2.4:1)

¹H NMR spectrum of **6** (major isomer)

NOESY spectrum of **6** (major isomer)

(12) Page 6 Ln 156: use “excellent” instead of “superb”.

Answer: As suggested by reviewer 2, We have changed “superb” to “excellent” in our revised manuscript.

(13) Page 7 Table 1: Is there any E/Z ratio or the double bond geometry is as indicated in the product? Indicate how double bond geometry was determined. If there is E/Z ratio indicate them for each product.

Answer: By comparing the crude NMR spectra as well as the spectra from the purified compound, we only detected the product **3a** with a single configuration.

¹H NMR spectrum of crude **3a**

¹H NMR spectrum of **3a**

Comparison of ^1H NMR spectrum of **3a** (blue) and crude **3a** (red)

The NOESY spectrum of **3a** suggests that product **3a** is in the *E* configuration.

NOESY spectrum of **3a**

In order to further confirm the configuration of **3a**, we obtained the single crystal structure of product *(R,R)*-**16**, providing additional evidence that product **3a** is in the *E* configuration.

For the other products listed in Table 1, we confirmed that only *E* configuration of product **3** was obtained through comparison of NMR spectra and NOESY spectra (e.g., **3g** and **3j**). We have included these results in our revised manuscript and supplementary information.

¹H NMR spectrum of crude **3g**

¹H NMR spectrum of **3g**

Comparison of ¹H NMR spectrum of **3g** (blue) and crude **3g** (red)

NOESY spectrum of **3g**

^1H NMR spectrum of crude **3j**

¹H NMR spectrum of **3j**

Comparison of ¹H NMR spectrum of **3j** (blue) and crude **3j** (red)

NOESY spectrum of **3j**

(14) For Table 1: Authors should include effect of sterics around carbon: (isopropyl, cyclohexyl and tert-butyl substitutions). The authors should also include results of aryl and heteroaryl substituents on carbon.

Answer: We appreciate Reviewer 2 for the favorable comments and helpful suggestions. We tested the effect of sterics around carbon; however, none of them could undergo the oxidative dienylation. We have included these results in our revised supplementary information (Table S5, Page S15).

Table S5: Survey of the effect of sterics around carbon for oxidative dienylation reactions^a

^aReactions were carried out with **S1** (0.1 mmol), **2a** (0.3 mmol), ⁿBu₄NPF₆ (0.15 mmol), quinuclidine (0.05 mmol), Ni(OAc)₂·4H₂O (10 mol%), **L9** (10 mol%), Cp₂Fe (10 mol%) and DCM/TFE = 2:1 (3 mL) at 25 °C in an undivided cell. TFE, 2,2,2-trifluoroethanol. nr = no reaction.

(15) It is interesting that the size of the ester makes a difference in the product ee. A comment in the text would be appropriate.

Answer: The size of the ester was varied (**3ab-3ae**), and the use of steric bulky benzoxazolyl acetate as the substrate led to an increase in the enantioselectivity of the reaction (**3ae**). We have included these results in our revised manuscript, as “The ester size was varied (**3ab-3ae**), and employing steric bulky benzoxazolyl acetate as the substrate resulted in enhanced enantioselectivity of the reaction (**3ae**).” We have also included these results in our revised supplementary information (Table S3, Page S14).

Table S3: Survey of esters^a

Reaction scheme showing the synthesis of product **3** from substrate **1** and **2a**. Reagents: Ni(OAc)₂·4H₂O (10 mol%), **L9** (10 mol%), Cp₂Fe, quinuclidine, ^tBu₄NPF₆, DCM/TFE, 25 °C, under Ar. Conditions: C(+)|Pt(-), 2.0 mA CCE, undivided cell.

Substrates **1** (S1f, S1g, S1h, **1a**, S1i) and product **3** (3ab, 3ac, 3ad, **3a**, 3ae) are shown.

Entry	Substrates 1	3	Yield (%)	e.e. (%) of 3
1	S1f	3ab	51	86
2	S1g	3ac	54	88
3	S1h	3ad	56	89
4	1a	3a	79	96
5	S1i	3ae	62	98

^aReactions were carried out with **1** (0.1 mmol), **2a** (0.3 mmol), ^tBu₄NPF₆ (0.15 mmol), quinuclidine (0.05 mmol), Ni(OAc)₂·4H₂O (10 mol%), **L9** (10 mol%), Cp₂Fe (10 mol%) and DCM/TFE = 2:1 (3 mL) at 25 °C in an undivided cell. TFE, 2,2,2-trifluoroethanol.

Methyl (*S,E*)-2-(benzo[d]oxazol-2-yl)-2-methyl-6-phenyl-3-(1-phenylvinyl)hex-3-enoate (**3ab**)

Reaction time: 3 h. ¹H NMR (400 MHz, CDCl₃) δ 7.73 – 7.63 (m, 1H), 7.54 – 7.46 (m, 1H), 7.37 – 7.30 (m, 2H), 7.29 – 7.26 (m, 1H), 7.25 – 7.24 (m, 1H), 7.23 – 7.18 (m, 2H), 7.19 – 7.12 (m, 4H), 7.11 – 7.04 (m, 2H), 5.83 (t, *J* = 7.3 Hz, 1H), 5.59 (d, *J* = 0.9 Hz, 1H), 4.91 (d, *J* = 1.1 Hz, 1H), 3.65 (s, 3H), 2.70 (t, *J* = 7.4 Hz, 2H), 2.52 – 2.38 (m, 2H), 1.76 (s, 3H). ¹³C NMR (100 MHz, CDCl₃) δ 172.21, 166.05, 150.94, 144.59, 141.55, 140.97, 139.22, 138.52, 133.24, 128.78, 128.33, 128.29, 127.70, 126.25, 125.97, 125.03, 124.24, 120.27, 117.20, 110.75, 55.24, 52.88, 35.82, 32.07, 22.84. ESI-MS: calculated [C₂₉H₂₇NO₃ + H]⁺: 438.2064, found: 438.2081. [α]_D²⁰ = +11.90 (*c* = 0.74, CH₂Cl₂). The product was analyzed by HPLC to determine the enantiomeric excess: 86% e.e. (CHIRALPAK IC, hexane/*i*-PrOH = 95/5, detector: 254 nm, T = 25 °C, flow rate: 1 mL/min), t₁(major) = 7.70 min, t₂(minor) = 16.78 min.

Ethyl (*S,E*)-2-(benzo[d]oxazol-2-yl)-2-methyl-6-phenyl-3-(1-phenylvinyl)hex-3-enoate (**3ac**)

Reaction time: 3 h. ¹H NMR (400 MHz, CDCl₃) δ 7.73 – 7.62 (m, 1H), 7.54 – 7.44 (m, 1H), 7.35 – 7.29 (m, 2H), 7.28 – 7.25 (m, 2H), 7.22 – 7.11 (m, 6H), 7.10 – 7.03 (m, 2H), 5.87 (t, *J* = 7.3 Hz, 1H), 5.59 (d, *J* = 1.0 Hz, 1H), 4.91 (d, *J* = 1.1 Hz, 1H), 4.13 (q, *J* = 7.1 Hz, 2H), 2.69 (t, *J* = 7.5 Hz, 2H), 2.48 – 2.35 (m, 2H), 1.75 (s, 3H), 1.18 (t, *J* = 7.1 Hz, 3H). ¹³C NMR (100 MHz, CDCl₃) δ 171.78, 166.25, 150.97, 144.62, 141.58, 141.00, 139.27, 138.48, 133.29, 128.74, 128.33, 128.28, 127.67, 126.24, 125.96, 124.96, 124.18,

120.24, 117.00, 110.70, 61.92, 55.18, 35.81, 32.07, 22.79, 14.06. **ESI-MS:** calculated $[C_{30}H_{29}NO_3 + H]^+$: 452.2220, found: 452.2228. $[\alpha]_D^{20} = +12.69$ ($c = 0.81$, CH_2Cl_2). The product was analyzed by HPLC to determine the enantiomeric excess: 88% e.e. (CHIRALPAK IC, hexane/*i*-PrOH =95/5, detector: 254 nm, T = 25 °C, flow rate: 1 mL/min), t_1 (major) = 7.02 min, t_2 (minor) = 16.20 min.

Isopropyl (*S,E*)-2-(benzo[d]oxazol-2-yl)-2-methyl-6-phenyl-3-(1-phenylvinyl)hex-3-enoate (3ad)

Reaction time: 3 h. **¹H NMR (400 MHz, CDCl₃)** δ 7.73 – 7.62 (m, 1H), 7.54 – 7.44 (m, 1H), 7.36 – 7.28 (m, 4H), 7.23 – 7.11 (m, 6H), 7.11 – 7.04 (m, 2H), 5.92 (t, $J = 7.3$ Hz, 1H), 5.60 (d, $J = 1.2$ Hz, 1H), 5.13 – 4.99 (m, 1H), 4.92 (d, $J = 0.8$ Hz, 1H), 2.69 (t, $J = 7.5$ Hz, 2H), 2.50 – 2.34 (m, 2H), 1.74 (s, 3H), 1.24 – 1.14 (m, 6H). **¹³C NMR (100 MHz, CDCl₃)** δ 171.38, 166.39, 150.98, 144.63, 141.60, 141.03, 139.34, 138.41, 133.37, 128.70, 128.33, 128.28, 127.66, 126.23, 125.94, 124.90, 124.13, 120.20, 116.76, 110.65, 69.61, 55.16, 35.80, 32.09, 22.65, 21.65, 21.57. **ESI-MS:** calculated $[C_{31}H_{31}NO_3 + H]^+$: 466.2377, found: 466.2393. $[\alpha]_D^{20} = +15.26$ ($c = 0.87$, CH_2Cl_2). The product was analyzed by HPLC to determine the enantiomeric excess: 89% e.e. (CHIRALPAK IC, hexane/*i*-PrOH =95/5, detector: 235 nm, T = 25 °C, flow rate: 1 mL/min), t_1 (major) = 5.88 min, t_2 (minor) = 13.21 min.

2-phenylpropan-2-yl

(*S,E*)-2-(benzo[d]oxazol-2-yl)-2-methyl-6-phenyl-3-(1-phenylvinyl)hex-3-enoate (3ae)

Reaction time: 3 h. **¹H NMR (400 MHz, CDCl₃)** δ 7.71 – 7.61 (m, 1H), 7.53 – 7.45 (m, 1H), 7.34 – 7.26 (m, 7H), 7.26 – 7.21 (m, 2H), 7.19 – 7.10 (m, 6H), 7.10 – 7.04 (m, 2H), 5.94 (t, $J = 7.3$ Hz, 1H), 5.57 (d, $J = 1.0$ Hz, 1H), 4.83 (d, $J = 0.9$ Hz, 1H), 2.68 (t, $J = 7.5$ Hz, 2H), 2.48 – 2.33 (m, 2H), 1.76 (s, 3H), 1.74 – 1.67 (m, 6H). **¹³C NMR (100 MHz, CDCl₃)** δ 170.27, 166.41, 151.02, 145.35, 144.53, 141.60, 141.07, 139.35, 138.37, 133.56, 128.70, 128.35, 128.32, 128.31, 127.69, 127.31, 126.21, 125.95, 124.91, 124.47, 124.14, 120.18, 116.52, 110.62, 83.62, 55.68, 35.79, 32.08, 28.19, 28.02, 22.63. **ESI-MS:** calculated $[C_{37}H_{35}NO_3 + H]^+$: 542.2690, found: 542.2703. $[\alpha]_D^{20} = +7.14$ ($c = 1.12$, CH_2Cl_2). The product was analyzed by HPLC to determine the enantiomeric excess: 98% e.e. (CHIRALPAK IC, hexane/*i*-PrOH =95/5, detector: 254 nm, T = 25 °C, flow rate: 1 mL/min), t_1 (major) = 5.44 min, t_2 (minor) = 7.04 min.

(16) Page 8 Table 2: products 8b-8e: what is X? Is it R instead of X?

Answer: The labeling is inconsistent. As suggested by reviewer 2, We have changed “R” to “X” in our revised manuscript.

(17) For Table 2: Authors should include effect of sterics around carbon: (isopropyl, cyclohexyl and tert-butyl substitutions). The authors should also include results of aryl and heteroaryl substituents on carbon.

Answer: We appreciate Reviewer 2 for the favorable comments and helpful suggestions. We tested the effect of sterics around carbon. We have included these results in our revised manuscript, as “However, the enantioselectivities exhibit a significant decrease when employing sterically substituted benzoxazolyl acetates (**8w** and **8x**). Further investigation revealed the compatibility of this method with various ester groups of benzoxazolyl acetates (**8y-8aa**).” We have also included these results in our revised supplementary information (Table S6, Page S15).

Table S6: Survey of the effect of sterics around carbon for oxidative allylation reactions^a

S1o

S1p

S1q

S1r

S1s

Entry	Substrates 1	8	Yield (%)	e.e. (%) of 8
1 ^b	S1o	8w	62	60
2 ^b	S1p	8x	67	61
3	S1q	—	nr	—
4	S1r	—	nr	—
5	S1s	—	nr	—

^aReactions were carried out with **S1** (0.1 mmol), **7a** (0.3 mmol), ^tBu₄NPF₆ (0.15 mmol), quinuclidine (0.05 mmol), Ni(OAc)₂·4H₂O (10 mol%), **L9** (10 mol%), Cp₂Fe (10 mol%) and DCM/TFE/MeOH = 3:2:1 (3 mL) at 25 °C in an undivided cell. ^b**7a** (0.6 mmol), 50 °C. TFE, 2,2,2-trifluoroethanol. nr = no reaction.

Ethyl (*R*)-2-(benzo[d]oxazol-2-yl)-2-isopropyl-4-phenylpent-4-enoate (**8w**)

Reaction time: 5 h. ¹H NMR (400 MHz, CDCl₃) δ 7.66 – 7.59 (m, 1H), 7.41 – 7.35 (m, 1H), 7.27 – 7.24 (m, 2H), 7.22 – 7.18 (m, 2H), 7.10 – 7.04 (m, 2H), 7.03 – 6.96 (m, 1H), 5.22 (d, *J* = 1.1 Hz, 1H), 5.16 (d, *J* = 1.2 Hz, 1H), 3.94 (q, *J* = 7.1 Hz, 2H), 3.51 (s, 2H), 2.75 – 2.62 (m, 1H), 1.14 (t, *J* = 7.1 Hz, 3H), 1.10 (d, *J* = 6.8 Hz, 3H), 0.97 (d, *J* = 6.8 Hz, 3H). ¹³C NMR (100 MHz, CDCl₃) δ 170.51, 165.19, 150.30, 144.62, 141.72, 140.57, 127.80, 127.01, 126.57, 124.71, 124.02, 120.03, 118.48, 110.56, 61.23, 57.46, 40.21, 32.77, 18.87, 18.70, 14.15. **ESI-MS:** calculated [C₂₃H₂₅NO₃ + H]⁺: 364.1907, found: 364.1921. [α]_D²⁰ = +4.69 (c = 0.51, CH₂Cl₂). The product was analyzed by HPLC to determine the enantiomeric excess: 60% e.e. (CHIRALPAK IC, hexane/*i*-PrOH = 95/5, detector: 254 nm, T = 25 °C, flow rate: 1 mL/min, t₁(major) = 5.48 min, t₂(minor) = 6.20 min.

Ethyl (*R*)-2-(benzo[d]oxazol-2-yl)-2-cyclohexyl-4-phenylpent-4-enoate (**8x**)

Reaction time: 5 h. ¹H NMR (400 MHz, CDCl₃) δ 7.69 – 7.61 (m, 1H), 7.43 – 7.36 (m, 1H), 7.28 – 7.23 (m, 4H), 7.16 – 7.09 (m, 2H), 7.09 – 7.01 (m, 1H), 5.20 (d, *J* = 1.2 Hz, 1H), 5.09 (d, *J* = 0.8 Hz, 1H), 4.03 – 3.88 (m, 2H), 3.59 – 3.45 (m, 2H), 2.31 – 2.18 (m, 1H), 2.07 – 1.90 (m, 2H), 1.79 – 1.63 (m, 2H), 1.61 – 1.54 (m, 1H), 1.24 – 1.10 (m, 5H), 1.10 – 0.94 (m, 2H), 0.82 – 0.67 (m, 1H). ¹³C NMR (100 MHz, CDCl₃) δ 170.48, 165.38, 150.29, 144.77, 142.02, 140.63, 127.87, 127.06, 126.66, 124.69, 124.04, 120.07, 118.25, 110.60, 61.16, 57.65, 43.25, 39.78, 29.16, 28.70, 26.90, 26.67, 26.46, 14.16. **ESI-MS:** calculated [C₂₆H₂₉NO₃ + H]⁺: 404.2220, found: 404.2229. [α]_D²⁰ = +3.75 (c = 0.58, CH₂Cl₂). The product was analyzed by HPLC to determine the enantiomeric excess: 61% e.e. (CHIRALPAK IC, hexane/*i*-PrOH = 95/5, detector: 254 nm, T = 25 °C, flow rate: 1 mL/min, t₁(major) = 5.04 min, t₂(minor) = 6.00 min.

(18) The authors have reported the leaving group effect for allylation reaction, but not for reactions with allenes. Why?

Answer: Screening different leaving groups of allylic reagents **7** suggested that allylmethylsilane gave the best results (Table 2). Then we also tested the leaving group effect of **2** for the dienylation reaction. Through the synthetic route below, we synthesized allenylmethyl phenyl sulfone **2m** and investigated its reaction behaviour in the oxidative dienylation. We have included the synthetic route in the revised supplementary information (Page S6)

(6-tosylhexa-3,4-diene-1,5-diy)dibenzene (2m)

¹H NMR (400 MHz, CDCl₃) δ 7.70 – 7.62 (m, 2H), 7.30 – 7.26 (m, 2H), 7.25 – 7.18 (m, 8H), 7.13 – 7.09 (m, 2H), 5.24 (t, *J* = 6.9 Hz, 1H), 4.12 (dd, *J* = 14.4, 0.7 Hz, 1H), 3.96 (dd, *J* = 14.4, 1.7 Hz, 1H), 2.65 (t, *J* = 7.3 Hz, 2H), 2.34 (s, 3H), 2.26 – 2.18 (m, 2H). **¹³C NMR (100 MHz, CDCl₃)** δ 208.51, 144.66, 141.10, 135.42, 134.36, 129.60, 128.90, 128.58, 128.51, 128.46, 127.10, 126.25, 126.18, 96.63, 94.23, 58.29, 35.11, 29.74, 21.65. **ESI-MS:** calculated [C₂₅H₂₄O₂S + Na]⁺: 411.1389, found: 411.1394.

Screening different leaving groups (LG) of allylic reagents **2** provided no improvement in the yield or enantioselectivity of the reaction. We have included these results in the revised supplementary information (Table S7, Page S16)

Table S7: Survey of the leaving group effect for oxidative dienylation reactions^a

Entry	Substrates 2	3	Yield (%)	e.e. (%) of 3
1	2a	3a	79	96
2	2m	3a	< 5	91
3	2n	3f	42	94

^aReactions were carried out with **1a** (0.1 mmol), **2** (0.3 mmol), ⁿBu₄NPF₆ (0.15 mmol), quinuclidine (0.05 mmol), Ni(OAc)₂·4H₂O (10 mol%), **L9** (10 mol%), Cp₂Fe (10 mol%) and DCM/TFE = 2:1 (3 mL) at 25 °C in an undivided cell. TFE, 2,2,2-trifluoroethanol.

(19) Benzoxazolyl acetates are reported in the study. Do other substituents such as amides, ketones, imides work?

Answer: As suggested by Reviewer 2, we have tried different kinds of carbonyl groups, including amides (**S1a**, **S1b**, **S1d**), imides **S1c** and ketone **S1e** for the reaction. As shown in Table S2 (Page S13 in our revised supplementary information), the use of different amine partners showed dramatic effect on the reactivity (entries 1-4). No reaction occurs in the presence of amides **S1a**, **S1b**, **S1c** (entries 1-3). Amide **S1d** proved to be suitable partner, delivering the desired product

3aa in 41% yield and 94% e.e. (entry 8). Ketone **S1e** exhibited good reactivity, resulting in the formation of the corresponding adduct **3z** in 59% yield with 78% enantioselectivity (entry 9). We have included these results in our revised manuscript (**3z** and **3aa**) and supplementary information (Page S13).

Table S2: Survey of substituent of benzoxazolyl acetates^a

Reaction conditions: Ni(OAc)₂·4H₂O (10 mol%), L9 (10 mol%), Cp₂Fe, quinuclidine, ^tBu₄NPF₆, DCM/TFE, 25 °C, under Ar, C(+) | Pt(-), 2.0 mA CCE, undivided cell.

Entry	Substrates S1	3	Yield (%)	e.e. (%) of 3
1	S1a	—	nr	—
2	S1b	—	nr	—
3	S1c	—	nr	—
4	S1d	3aa	36	92
5 ^b	S1d	3aa	42	92
6 ^c	S1d	3aa	37	92
7 ^d	S1d	3aa	nr	—
8 ^e	S1d	3aa	41	94
9 ^e	S1e	3z	59	78

^aReactions were carried out with **S1** (0.1 mmol), **2a** (0.3 mmol), ^tBu₄NPF₆ (0.15 mmol), quinuclidine (0.05 mmol), Ni(OAc)₂·4H₂O (10 mol%), **L9** (10 mol%), Cp₂Fe (10 mol%) and DCM/TFE = 2:1 (3 mL) at 25 °C in an undivided cell. ^b**2a** (0.6 mmol). ^c**2a** (0.6 mmol), 50 °C. ^d**2a** (0.6 mmol), HCOOH (20 mol%), Ni(OAc)₂·4H₂O (20 mol%), **L9** (20 mol%), **2a** (0.6 mmol), 50 °C. TFE, 2,2,2-trifluoroethanol. nr = no reaction.

(S,E)-2-(benzo[d]oxazol-2-yl)-N,2-dimethyl-6-phenyl-3-(1-phenylvinyl)hex-3-enamide (3aa)

Reaction time: 3 h. ¹H NMR (400 MHz, CDCl₃) δ 8.39 (q, *J* = 4.1 Hz, 1H), 7.62 – 7.52 (m, 1H), 7.42 – 7.35 (m, 1H), 7.32 – 7.26 (m, 3H), 7.25 – 7.20 (m, 2H), 7.19 – 7.14 (m, 1H), 7.14 – 7.08 (m, 2H), 7.06 – 7.00 (m, 4H), 5.94 (t, *J* = 7.2 Hz, 1H), 5.41 (d, *J* = 1.4 Hz, 1H), 4.68 (d, *J* = 1.5 Hz, 1H), 2.82 – 2.65 (m, 5H), 2.49 – 2.37 (m, 2H), 1.82 (s, 3H). ¹³C NMR (100 MHz, CDCl₃) δ 170.26, 166.98, 149.98, 145.09, 141.62, 140.62, 140.04, 138.92, 131.25, 128.82, 128.38, 127.96, 127.38, 126.10, 126.01, 125.11, 124.39, 119.78, 117.11, 110.66, 54.29, 35.93, 31.93, 26.66, 22.23. **ESI-MS**: calculated [C₂₉H₂₈N₂O₂ + H]⁺: 437.2224, found: 437.2228. [α]_D²⁰ = -20.80 (c = 0.52, CH₂Cl₂). The product was analyzed by HPLC to determine the enantiomeric excess: 94% e.e. (CHIRALPAK IE, hexane/*i*-PrOH = 80/20, detector: 254 nm, T = 25 °C, flow rate: 1 mL/min), t₁(major) = 8.80 min, t₂(minor) = 9.93 min.

(S,E)-2-(benzo[d]oxazol-2-yl)-2-methyl-1,6-diphenyl-3-(1-phenylvinyl)hex-3-en-1-one (3z)

Reaction time: 3 h. ¹H NMR (400 MHz, CDCl₃) δ 7.88 – 7.83 (m, 2H), 7.62 – 7.58 (m, 1H), 7.51 – 7.45 (m, 2H), 7.40 – 7.34 (m, 2H), 7.34 – 7.27 (m, 2H), 7.21 – 7.13 (m, 3H), 7.11 – 7.07 (m, 2H), 7.07 – 7.03 (m, 3H), 7.02 – 6.97 (m, 2H), 6.06 (t, *J* = 7.2 Hz, 1H), 5.59 (d, *J* = 1.3 Hz, 1H), 5.03 (d, *J* = 1.1 Hz, 1H), 2.77 – 2.66 (m, 2H), 2.62 – 2.50 (m, 2H), 1.82 (s, 3H). ¹³C NMR (100 MHz, CDCl₃) δ 198.60, 166.78, 150.91, 145.35, 141.54, 141.08, 139.54, 138.28, 135.93, 135.56, 132.56, 130.06, 128.70, 128.42, 128.24, 127.72,

126.40, 126.02, 124.88, 124.16, 120.15, 117.89, 110.76, 59.59, 35.80, 32.52, 22.99. **ESI-MS:** calculated $[C_{34}H_{29}NO_2 + H]^+$: 484.2271, found: 484.2285. $[\alpha]^{20}_D = +33.40$ ($c = 0.59$, CH_2Cl_2). The product was analyzed by HPLC to determine the enantiomeric excess: 78% e.e. (CHIRALPAK IC, hexane/*i*-PrOH =80/20, detector: 254 nm, T = 25 °C, flow rate: 1 mL/min), t_1 (major) = 4.87 min, t_2 (minor) = 9.52 min.

(20) Does benzimidazole or benzothiazole auxiliary work? How about a simple oxazole in place of benzoxazole?

Answer: We appreciate Reviewer 2 for the favorable comments and helpful suggestions. As suggested by reviewer 2, we have investigated different auxiliary in place of benzoxazole for the reaction. As shown in Table S4, varying the substrate **1** with different substituents from benzoxazole to benzimidazole **S1j**, benzothiazole **S1k**, 2-pyridyl **S1l**, and dihydrothiazol **S1m**, failed to participate the reaction (entries 1-4). The corresponding product was isolated in 56% yield and 28% e.e. with simple oxazole **S1n** as substrate (entry 5). We have included these results in our revised supplementary information (Table S4, page S14).

Table S4: Survey of benzoxazole^a

Reaction scheme showing the conversion of substrate **S1** and alkene **2a** to product **3**. Conditions: Ni(OAc)₂·4H₂O (10 mol%), **L9** (10 mol%), Cp₂Fe, quinuclidine, ^tBu₄NPF₆, DCM/TFE, 25 °C, under Ar, C(+)|Pt(-), 2.0 mA CCE, undivided cell.

Substrates **S1j**, **S1k**, **S1l**, **S1m**, and **S1n** are shown below the reaction scheme.

Entry	Substrates S1	3	Yield (%)	e.e. (%) of 3
1	S1j	—	nr	—
2	S1k	—	nr	—
3	S1l	—	nr	—
4	S1m	—	nr	—
5	S1n	3af	56	28

^aReactions were carried out with **S1** (0.1 mmol), **2a** (0.3 mmol), ^tBu₄NPF₆ (0.15 mmol), quinuclidine (0.05 mmol), Ni(OAc)₂·4H₂O (10 mol%), **L9** (10 mol%), Cp₂Fe (10 mol%) and DCM/TFE = 2:1 (3 mL) at 25 °C in an undivided cell. TFE, 2,2,2-trifluoroethanol. nr = no reaction.

Ethyl (*S,E*)-2-(4,5-dihydrooxazol-2-yl)-2-methyl-6-phenyl-3-(1-phenylvinyl)hex-3-enoate (**3af**)

Reaction time: 4 h. ¹H NMR (400 MHz, CDCl₃) δ 7.37 – 7.31 (m, 2H), 7.28 – 7.21 (m, 5H), 7.20 – 7.14 (m, 1H), 7.13 – 7.07 (m, 2H), 5.84 (t, *J* = 7.3 Hz, 1H), 5.62 (d, *J* = 1.5 Hz, 1H), 5.02 (d, *J* = 1.5 Hz, 1H), 4.20 – 4.06 (m, 3H), 4.06 – 3.94 (m, 1H), 3.80 – 3.68 (m, 1H), 3.66 – 3.53 (m, 1H), 2.75 – 2.62 (m, 2H), 2.45 – 2.34 (m, 2H), 1.48 (s, 3H), 1.21 (t, *J* = 7.1 Hz, 3H). ¹³C NMR (100 MHz, CDCl₃) δ 172.09, 168.08, 144.95, 141.78, 139.68, 137.82, 132.52, 128.75, 128.32, 128.27, 127.67, 126.41, 125.93, 116.74, 67.72, 61.56, 54.37, 54.24, 35.88, 31.97, 22.40, 14.08. **ESI-MS:** calculated $[C_{26}H_{29}NO_3 + Na]^+$: 426.2040, found: 426.2054. $[\alpha]^{20}_D = +8.97$ ($c = 1.00$, CH_2Cl_2). The product was analyzed by HPLC to determine the enantiomeric excess: 28% e.e. (CHIRALPAK IC, hexane/*i*-PrOH =80/20, detector: 254 nm, T = 25 °C, flow rate: 1 mL/min), t_1 (major) = 8.38 min, t_2 (minor) = 10.55 min.

Reply to comments by Reviewer 3

We appreciate Reviewer 3 for the favorable comments and helpful suggestions!

- (1) The abstract and the introduction give the impression that there is little precedent in enantioselective Lewis acid catalysis with radical intermediates. In fact, this is an established strategy. The introduction should mention the key reports in this area so that the reader can properly contextualize the study. For example: Huang, X.; Zhang, Q.; Lin, J.; Harms, K.; Meggers, E., Electricity-driven asymmetric Lewis acid catalysis. *Nat. Catal.* 2019, 2 (1), 34–40., Zhang, Q.; Liang, K.; Guo, C., Enantioselective Nickel-Catalyzed Electrochemical Radical Allylation. *Angew. Chem. Int. Ed.* 2022, 61 (38), e202210632.

Answer: We appreciate Reviewer 3 for the favorable comments and helpful suggestions! Those comments are greatly valuable and helpful for revising and improving our paper. We have made all the necessary amendments as suggested in our revised manuscript and revised supplementary information. We have included “Recently, Meggers^{39,40} and our group⁴¹⁻⁴³ showed the use of Lewis acid catalysis in asymmetric radical electrochemical transformations of 2-acyl imidazoles.” in the introduction of the revised manuscript.

- (2) In the abstract: The term "ingenious" may be somewhat overly enthusiastic. A more tempered expression should be chosen.

Answer: As suggested by reviewer 3, We have changed “ingenious” to “thoughtful” in our revised manuscript.

- (3) In line 50: "Incredibly low potential" is not a scientific expression. This could be rephrased.

Answer: As suggested by reviewer 3, We have changed “Incredibly low potential” to “the coordination of the catalyst with the substrate results in a decreased oxidation potential” in our revised manuscript.

- (4) In line 62: Typo: “anode oxidation conditions”

Answer: As suggested by reviewer 3, We have corrected this errors in our revised manuscript: “anode oxidation conditions” was changed to “anodic oxidation conditions”.

- (5) In line 63: "Exceptional yields". In conclusion: “remarkable yields” These are not suitable descriptions. The yields are more in the range of 50-60%, and yields over 90% are not achieved.

Answer: As suggested by reviewer 3, We have changed “Exceptional yields” and “remarkable yields” to “good yields” in our revised manuscript.

- (6) Caption of Fig. 1: “D” should be “d”

Answer: As suggested by reviewer 3, We have changed “D” to “d” in our revised manuscript.

(7) The gram-scale reaction should be discussed more in the context of synthetic utility rather than within the mechanistic studies.

Answer: As suggested by Reviewer 3, we have moved the gram-scale reaction to the section of synthetic utility as Fig. 3b.

(8) Neither for the substrate scope nor the gram-scale reaction times are provided. However, this is a crucial reaction parameter for reproducing the reaction and determining the energy efficiency (Faradaic yield).

Answer: As suggested by reviewer 3, We have provided reaction times in our revised manuscript and revised supplementary information.

(9) In the catalytic cycle, no specific information is provided regarding the fate of the TMS group. What do the authors suggest? Is it released as a TMS cation or as a TMS radical? Is TMA-OCH₂CF₃ obtained as a stoichiometric byproduct? If the TMS radical is released, anodic oxidation would not be immediately necessary. The authors should provide an explanation here.

Answer: We appreciate Reviewer 3 for the favorable comments and helpful suggestions. Through crude ¹H NMR and high-resolution mass spectrometry detection, TMS-OCH₂CF₃ was identified as a stoichiometric byproduct.

Tracking and monitoring experiment of TMS-OCH₂CF₃

In a dried sealed tube, Ni(OAc)₂·4H₂O (12.4 mg, 0.05 mmol) and **L9** (18.0 mg, 0.05 mmol) were dissolved in CD₂Cl₂ (3.0 mL) under N₂ atmosphere, and the mixture was stirred for 1 h at room temperature before use. A 30 mL flask equipped with a magnetic stir bar was charged with **1a** (124.5 mg, 0.5 mmol), **2a** (455 mg, 1.5 mmol), ⁿBu₄NPF₆ (290 mg, 0.75 mmol), quinuclidine (28 mg, 0.25 mmol), Cp₂Fe (9.3 mg, 0.05 mmol) and 1,1,2,2-Tetrachloroethane (42 mg, 0.25 mmol, as an internal standard). The flask was equipped with a carbon plate (2.0 cm × 2.0 cm × 2.0 mm) as the anode and a platinum plate (2.0 cm × 2.0 cm × 0.2 mm) as the cathode. The reaction mixture was degassed via vacuum evacuation and backfilled with argon three times, followed by the addition via a syringe of nickel catalyst solution made in advance, CD₂Cl₂ (7.0 mL) and TFE (5.0 mL). The constant current (8.0 mA) electrolysis was carried out at 25 °C. A reaction solution (0.5 mL) was taken every half an hour and analyzed by ¹H NMR spectroscopy. Yields determined by ¹H NMR using 1,1,2,2-Tetrachloroethane as an internal standard.

^1H NMR (400 MHz, $\text{CD}_2\text{Cl}_2:\text{TFE} = 2:1$) of reaction mixture

^1H NMR spectrum of $\text{TMS-OCH}_2\text{CF}_3$ (after purification)

^1H NMR (400 MHz, CD_2Cl_2) δ 3.93 (q, $J = 8.8$ Hz, 1H), 0.17 (s, 1H).

High resolution mass spectrometry (HRMS) of reaction mixture

EI-MS: calculated $[\text{C}_5\text{H}_{11}\text{F}_3\text{OSi}]^+$: 172.0526, found: 172.0518

Additionally, we investigated the process of removing TMS from the radical intermediate through DFT calculations. We considered two possible reaction pathways: Path 1) TMS leaves as a radical and forms the product, followed by TMS radical undergoing electrode oxidation and combining with TFE (red pathway). Path 2) The radical intermediate undergoes oxidation to form a cation, followed by nucleophilic attack of TFE on TMS to generate the product (black pathway). The process through Path 1 is thermodynamically unfavorable ($\Delta G = 25.4$ kcal/mol), suggesting that Path 1 is unlikely to occur at room temperature. In Path 2, we calculated the process of oxidation of the radical intermediate at a potential of +0.4 V (versus SCE) and found that this process is energetically favorable ($\Delta G = -17.6$ kcal/mol). Subsequently, the nucleophilic attack of TFE on TMS has a low energy barrier ($\Delta G = 12.4$ kcal/mol) and is relatively facile. Therefore, we suggested that the radical intermediate is initially oxidized to a cation and then undergoes TMS removal to yield the target product. We have included these results in our revised manuscript and supplementary information (Page S58- Page S66).

Figure S11. Gibbs free energy profiles of TMS leaving. Computed at the M06-2X-D3(0)/def2-TZVP/SMD(DCM)//B3LYP-D3(BJ)/def2-SVP/IEFPCM(DCM) level. Paths with an electricity icon refers to an electrochemical process under the external potential of 0.4 V (versus SCE). The 'barrierless' refers to the absence of electronic barrier that makes the transition state coordinates difficult to locate.

(10) To determine the role of ferrocene, it would be interesting to test whether $\text{Cp}_2\text{Fe}^{\text{III}}(\text{PF}_6)$ is effective as a stoichiometric oxidizing agent for the reaction.

Answer: To better understand the process of ferrocene-mediated oxidation, we performed CV tests that revealed a rise in ferrocene's oxidation peak and a drop in its reduction peak when catalyst concentrations increased. This implies that an electron transfer occurred between the oxidized ferrocene [$\text{Cp}_2\text{Fe}^{\text{III}}(\text{PF}_6)$] and the enolate intermediate.

Figure S7. Titration of Ni/(*R,R*)-**L9** (3.3 mM to 33 mM) to Cp_2Fe (1.5 mM) monitored by cyclic voltammetry in the presence of **1a** (33 mM), quinuclidine (17 mM), ${}^n\text{Bu}_4\text{NPF}_6$ (50 mM) in DCM/TFE = 2:1 (3 mL). A decrease in ferrocene's reduction peak were observed with increasing concentrations of Ni/(*R,R*)-**L9**. We have included these results in our revised supplementary information (Page S47, Figure S7)

Using stoichiometric $\text{Cp}_2\text{Fe}^{\text{III}}(\text{PF}_6)$ instead of the electrochemical oxidation process only afford **3a** in 17% yield and an 84% e.e. The low yield may be attributed to the unfavorable oxidizing step of the radical intermediate to remove TMS.

(11) The role of quinuclidine is not explicitly explained. Is it possible that it serves as a HAT (Hydrogen Atom Transfer) mediator and is involved in the H-abstraction? The authors should clarify their position on this.

Answer: As suggested by reviewer 3, we investigated the use of different types of bases in the reaction (Table S10). The reaction proceeded even without the addition of a base, affording **3a** in 51% and 96% e.e. (entry 1). Other organic bases such as 2,6-dimethylpyridine, triethylamine, or DMAP also yielded reaction outcomes similar to quinuclidine (entries 2-4 vs 5). Inorganic bases also effectively promoted the reaction, resulting **3a** in 73% and 95% e.e. (entry 6). These results indicate that various types of bases can effectively promote the reaction. We have included these results in our revised supplementary information (Page S17, Table S10).

Table S10: Survey of base^a

Entry	Base	Yield (%)	e.e. (%) of 3a
1	No base	51	96
2	2,6-dimethoxyypyridine	49	96
3	Et ₃ N	74	96
4	DMAP	57	95
5	quinuclidine	79	96
6	K ₂ CO ₃	73	95

^aReactions were carried out with **1a** (0.1 mmol), **2a** (0.3 mmol), ^tBu₄NPF₆ (0.15 mmol), base (0.05 mmol), Ni(OAc)₂·4H₂O (10 mol%), **L9** (10 mol%), Cp₂Fe (10 mol%) and DCM/TFE = 2:1 (3 mL) at 25 °C in an undivided cell. TFE, 2,2,2-trifluoroethanol.

Constant potential experiment:

The onset potential of [**1a** + quinuclidine + Ni/(*R,R*)-**L9**] was measured approximately at +0.36 V (versus SCE), and the onset potential of quinuclidine was measured approximately at +1.3 V (versus SCE). When a controlled potential experiment was conducted with a constant anodic potential of +0.4 V (versus SCE) at room temperature, which is higher than the potential required for the oxidation of nickel-bound enolate but insufficient for the direct oxidation of quinuclidine (approximately +1.3 V versus SCE), we observed the formation of the product **3a** in 75% yield and 96% e.e. All these results indicate that quinuclidine primarily acts as a base in this reaction. We have included these results in our revised supplementary information (Page S53)

(12) In Fig. 2j: $\frac{1}{2}$ H₂

Answer: As suggested by reviewer 3, We have changed “H₂” to “ $\frac{1}{2}$ H₂” in our revised manuscript.

(13) In line 156: "Superb" is a too positive term for the actual yields obtained.

Answer: As suggested by reviewer 3, We have changed “superb” to “excellent” in our revised manuscript.

(14) Fig 3b and 3c: Typo “Plenylglycinol”

Answer: As suggested by reviewer 3, We have corrected this errors in our revised manuscript: “Plenylglycinol” was changed to “Phenylglycinol”.

(15) What are the d.r. values for the synthesized Boox ligands? Since the nitriles (compound 15 and 18) were not used with >99% ee, diastereomers should be obtained. Could these be separated chromatographically? If so, the Boox ligands would likely be more easily accessible even from the racemic nitriles.

Answer: We appreciate Reviewer 3 for the favorable comments and helpful suggestions. We detected the diastereoselectivity (d.r.) of target products **16** and **19** by ¹H NMR, and also checked the enantioselectivity of starting materials **15** and **18**. It was found that the enantioselectivity of recovered starting materials **15** or **18** is related to the diastereoselectivity of products **16** and **19**. Through thin-layer chromatography (TLC) detection, we found that the polarities of the two isomers of **16** and **19** are very close, making it difficult to purify diastereomeric isomers through column chromatography separation. We have included these results in our revised manuscript and supplementary information (Page S39-S42)

1H NMR spectrum of (R,R)-16 (74:1 dr)

¹H NMR spectrum of (*R,S*)-16 (21:1 dr)

¹H NMR spectrum of (*R,R*)-19 (29:1 dr)

¹H NMR spectrum of (*R,S*)-**19** (18:1 dr)

(16) In conclusion: “outstanding performance” is too positive. The yields and ee values are not that high.

Answer: As suggested by reviewer 3, We have changed “outstanding performance” to “efficient” in our revised manuscript.

(17) Supporting Information. Page S12 and S13: No reaction times for the general procedures are provided

Answer: As suggested by reviewer 3, We have provided reaction times in our revised manuscript and supplementary information (Page S19).

(18) Supporting Information. Page S26: No reaction times for the gram scale is provided

Answer: As suggested by reviewer 3, We have provided reaction times in our revised supplementary information (Page S35).

(19) Supporting Information. Page S30 and S31: Typo “Plenylglycinol”, “Plenylglycinolin”

Answer: As suggested by reviewer 3, We have corrected these errors in our revised supplementary information: “Plenylglycinol” was changed to “Phenylglycinol”; “Plenylglycinolin” was changed to “Phenylglycinol”.

(20) Supporting Information. Page S47 and S48: The Flack parameters are not mentioned, although they are crucial for determining the absolute configurations.

Answer: As suggested by Reviewer 3, we have included the Flack parameters in our revised supplementary information (Page S67-S69). The Flack parameters for **3f**, **8l**, and (*R,R*)-**16** are

0.03(9), 0.00(7), and 0.05(13), respectively. These parameters are very close to zero, indicating that the absolute configurations of the products are correct.

Reviewers' Comments:

Reviewer #1:

Remarks to the Author:

The authors have made suitable revisions according to the referee comments. The manuscript can be accepted.

Reviewer #2:

Remarks to the Author:

Chiral nickel catalyzed dienylation and allylation of benzooxazolyl acetates under anodic oxidative conditions are reported in this communication. The reaction showed good scope with respect to benzooxazolyl acetates, allenyl silanes and allyl silanes. Through CV and control studies, a radical mechanism is proposed. The authors have answered this reviewers comments and the manuscript can be recommended for publication after briefly considering the following.

Comments/Questions:

Page 6: use "sterically demanding" instead of "steric bulky"

Figure 1d: what is the use of quinuclidine?

What happens in the case of α -unsubstituted benzooxazolyl acetate (when R=H)?

Some of the NOESY spectra are not processed well in both F2 and F1 direction (NOESY of 3g for instance).

Are the starting materials s1o-s1s known? If not include the characterization information, The authors should check with other starting materials that have been included after the revision for spectral data.

Reviewer #3:

Remarks to the Author:

The authors have done an excellent job and have addressed my concerns as well as those of the other reviewers convincingly. The manuscript can be published as it is.

Responses to Comments

for

Enantioselective Nickel-Catalyzed Anodic Oxidative Dienylation and Allylation Reactions

Qinglin Zhang, Jiayin Zhang, Wangjie Zhu, Ruimin Lu, Chang Guo*

Please find below a list of comments and changes made to the above manuscript in response to reviewers. Further changes made to the manuscript since submission are also listed at the end of this document. **A copy of the revised manuscript, a word document showing tracked changes** made to the manuscript since submission, and **revised supplementary information** are also included as part of this revision.

Reply to comments by Reviewer 1

- (1) The authors have made suitable revisions according to the referee comments. The manuscript can be accepted.

Answer: We appreciate Reviewer 1 for the favorable comments.

Reply to comments by Reviewer 2

We appreciate Reviewer 2 for the favorable comments and many helpful suggestions!

- (1) Chiral nickel catalyzed dienylation and allylation of benzooxazolyl acetates under anodic oxidative conditions are reported in this communication. The reaction showed good scope with respect to benzooxazolyl acetates, allenyl silanes and allyl silanes. Through CV and control studies, a radical mechanism is proposed. The authors have answered this reviewers comments and the manuscript can be recommended for publication after briefly considering the following.

Answer: We appreciate Reviewer 2 for the favorable comments and helpful suggestions! Those comments are greatly valuable and helpful for revising and improving our paper. We have made all the necessary amendments as suggested in our revised manuscript and revised supplementary information.

- (2) Page 6: use “sterically demanding” instead of “steric bulky”

Answer: As suggested by reviewer 2, We have changed “steric bulky” to “sterically demanding” in our revised manuscript.

(3) Figure 1d: what is the use of quinuclidine?

Answer: As suggested by reviewer 2, we investigated the use of different types of bases in the reaction (Table S10). The reaction proceeded even without the addition of a base, affording **3a** in 51% and 96% e.e. (entry 1). Other organic bases such as 2,6-dimethylpyridine, triethylamine, or DMAP also yielded reaction outcomes similar to quinuclidine (entries 2-4 vs 5). Inorganic bases also effectively promoted the reaction, resulting **3a** in 73% and 95% e.e. (entry 6). These results indicate that various types of bases can effectively promote the reaction. We have included these results in our revised supplementary information (Page S21, Table S10).

Table S10: Survey of base^a

Entry	Base	Yield (%)	e.e. (%) of 3a
1	No base	51	96
2	2,6-dimethoxyypyridine	49	96
3	Et ₃ N	74	96
4	DMAP	57	95
5	quinuclidine	79	96
6	K ₂ CO ₃	73	95

^aReactions were carried out with **1a** (0.1 mmol), **2a** (0.3 mmol), ^tBu₄NPF₆ (0.15 mmol), base (0.05 mmol), Ni(OAc)₂·4H₂O (10 mol%), **L9** (10 mol%), Cp₂Fe (10 mol%) and DCM/TFE = 2:1 (3 mL) at 25 °C in an undivided cell. TFE, 2,2,2-trifluoroethanol.

Constant potential experiment:

The onset potential of [**1a** + quinuclidine + Ni/(*R,R*)-**L9**] was measured approximately at +0.36 V (*versus* SCE). The onset potential of quinuclidine was measured approximately at +1.3 V (*versus* SCE). When a controlled potential experiment was conducted with a constant anodic potential of +0.4 V (*versus* SCE) at room temperature, which is higher than the potential required for the oxidation of nickel-bound enolate but insufficient for the direct oxidation of quinuclidine (approximately +1.3 V *versus* SCE), we observed the formation of the product **3a** in 75% yield and 96% e.e. All these results indicate that quinuclidine primarily acts as a base in this reaction. We have included these results in our revised supplementary information (Page S57)

(4) What happens in the case of unsubstituted benzooxazolyl acetate (when R=H)?

Answer: As suggested by reviewer 2, we attempted to use unsubstituted benzooxazolyl acetate as a substrate under standard conditions. However, only the corresponding racemic product was obtained in 36% yield and >20:1 (Z/E). We have included this result in our revised supplementary information (Page S19, Table S5).

Tert-butyl (Z)-2-(benzo[d]oxazol-2-yl)-6-phenyl-3-(1-phenylvinyl)hex-3-enoate (3ag) (Z/E >20:1)

Reaction time: 3 h. ¹H NMR (400 MHz, CDCl₃) δ 7.76 – 7.69 (m, 1H), 7.56 – 7.48 (m, 1H), 7.37 – 7.30 (m, 4H), 7.29 – 7.24 (m, 3H), 7.20 – 7.11 (m, 3H), 7.08 – 7.01 (m, 2H), 5.94 (t, *J* = 7.2 Hz, 1H), 5.56 (d, *J* = 1.2 Hz, 1H), 5.03 (d, *J* = 1.1 Hz, 1H), 4.62 (s, 1H), 2.75 – 2.61 (m, 2H), 2.55 – 2.40 (m, 2H), 1.43 (s, 9H). ¹³C NMR (100 MHz, CDCl₃) δ 167.50, 162.39, 151.14, 145.62, 141.52, 141.34, 138.28, 134.57, 133.61, 128.68, 128.57, 128.36, 128.06, 126.78, 125.94, 124.99, 124.30, 120.36, 116.42, 110.75, 82.50, 52.86, 35.82, 31.50, 27.99. **ESI-MS:** calculated [C₃₁H₃₁NO₃ + H]⁺: 466.2377, found: 466.2375. The product was analyzed by HPLC to determine the enantiomeric excess: racemic (CHIRALPAK IE, hexane/*i*-PrOH = 95/5, detector: 254 nm, T = 25 °C, flow rate: 1 mL/min), t₁ = 8.19 min, t₂ = 11.80 min.

¹H NMR spectrum of crude 3ag (Z/E = 25:1)

^1H NMR spectrum of **3ag**

^{13}C NMR spectrum of **3ag**

NOESY spectrum of **3ag**

Rac-3ag

SAMPLE INFORMATION			
Sample Name:	zql-17-287-rac-2-IE-5%	Acquired By:	System
Sample Type:	Unknown	Sample Set Name:	0
Vial:	81	Acq. Method Set:	5%210400
Injection #:	1	Processing Method:	17287
Injection Volume:	10.00 ul	Channel Name:	254.0nm
Run Time:	16.0 Minutes	Proc. Chnl. Descr.:	2998 PDA 254.0 nm (2998)
Date Acquired:	4/30/2024 9:45:47 PM CST		
Date Processed:	5/1/2024 11:24:34 AM CST		

	RT	Area	% Area	Height
1	8.172	430566	49.59	35049
2	11.785	437673	50.41	24670

Asy-3ag

SAMPLE INFORMATION			
Sample Name:	zql-17-287-123-2-IE-5%	Acquired By:	System
Sample Type:	Unknown	Sample Set Name:	0
Vial:	82	Acq. Method Set:	5%210400
Injection #:	1	Processing Method:	172872
Injection Volume:	10.00 ul	Channel Name:	254.0nm
Run Time:	16.0 Minutes	Proc. Chnl. Descr.:	2998 PDA 254.0 nm (2998)
Date Acquired:	4/30/2024 10:02:28 PM CST		
Date Processed:	5/1/2024 11:25:28 AM CST		

	RT	Area	% Area	Height
1	8.192	1836683	50.06	165459
2	11.804	1832351	49.94	112149

(5) Some of the NOESY spectra are not processed well in both F2 and F1 direction (NOESY of 3g for instance).

Answer: As suggested by reviewer 2, We have reprocessed all the NOESY spectra as follows.

NOESY spectrum of **3a**

NOESY spectrum of **3g**

NOESY spectrum of **3j**

NOESY spectrum of **5**

NOESY spectrum of **6**

(6) Are the starting materials s1o-s1s known? If not include the characterization information, The authors should check with other starting materials that have been included after the revision for spectral data.

Answer: As suggested by reviewer 2, we have included the characterization information of starting materials in our revised supplementary information (Page S2- Page S7). As shown below:

Preparation of benzoxazolyl acetates S1c, S1o-S1s¹⁻³

To a solution of NaHMDS (10 mL, 2 M in THF, 20 mmol, 2.0 equiv.) in dry toluene (20 mL) at 0 °C was added amide or ester (20 mmol, 2.0 equiv.) dropwise. After being stirred at 0 °C for 30 minutes, 2-chlorobenzoxazole (1.54 g, 10 mmol, 1.0 equiv.) was added at 0 °C. The reaction mixture was allowed to warm to room temperature and stirred for 3 hours. After completion, saturated aqueous NH₄Cl was added and the aqueous layer was extracted with EtOAc (20 mL × 3). The combined organic layers were washed with brine, dried over Na₂SO₄, filtered and concentrated in vacuo. The crude product was purified by flash chromatography to afford the desired product **S1c, S1o-S1s**.

Tert-butyl (2-(benzo[d]oxazol-2-yl)propanoyl)(methyl)carbamate (S1c)

¹H NMR (400 MHz, CDCl₃) δ 7.76 – 7.64 (m, 1H), 7.56 – 7.45 (m, 1H), 7.35 – 7.22 (m, 2H), 5.25 (q, *J* = 7.0 Hz, 1H), 3.25 (s, 3H), 1.72 (d, *J* = 7.1 Hz, 3H), 1.43 (s, 9H). ¹³C NMR (100 MHz, CDCl₃) δ 173.08, 165.75, 153.06, 150.86, 141.19, 124.81, 124.27, 120.01, 110.60, 83.81, 42.16, 32.25, 27.94, 16.02. **ESI-MS: calculated** [C₁₆H₂₀N₂O₄ + H]⁺: 305.1496, **found**: 305.1497.

Ethyl 2-(benzo[d]oxazol-2-yl)-3-methylbutanoate (S1o)

$^1\text{H NMR}$ (400 MHz, CDCl_3) δ 7.79 – 7.65 (m, 1H), 7.61 – 7.48 (m, 1H), 7.43 – 7.30 (m, 2H), 4.32 – 4.12 (m, 2H), 3.80 (d, $J = 9.0$ Hz, 1H), 2.80 – 2.65 (m, 1H), 1.26 (t, $J = 7.1$ Hz, 3H), 1.11 (d, $J = 6.7$ Hz, 3H), 1.00 (d, $J = 6.7$ Hz, 3H). $^{13}\text{C NMR}$ (100 MHz, CDCl_3) δ 169.39, 162.89, 150.96, 141.09, 125.12, 124.46, 120.23, 110.80, 61.65, 53.75, 30.36, 20.85, 20.46, 14.24. **ESI-MS: calculated** $[\text{C}_{14}\text{H}_{17}\text{NO}_3 + \text{H}]^+$: 248.1281, **found**: 248.1281.

Ethyl 2-(benzo[d]oxazol-2-yl)-2-cyclohexylacetate (S1p)

$^1\text{H NMR}$ (400 MHz, CDCl_3) δ 7.79 – 7.66 (m, 1H), 7.59 – 7.48 (m, 1H), 7.38 – 7.27 (m, 2H), 4.29 – 4.12 (m, 2H), 3.82 (d, $J = 9.5$ Hz, 1H), 2.46 – 2.31 (m, 1H), 1.92 – 1.82 (m, 1H), 1.80 – 1.73 (m, 1H), 1.72 – 1.60 (m, 2H), 1.60 – 1.50 (m, 1H), 1.42 – 1.27 (m, 2H), 1.24 (t, $J = 7.1$ Hz, 3H), 1.21 – 1.03 (m, 3H). $^{13}\text{C NMR}$ (100 MHz, CDCl_3) δ 169.32, 162.77, 150.98, 141.12, 125.08, 124.45, 120.21, 110.82, 61.64, 53.00, 39.45, 31.29, 30.75, 26.10, 26.02, 25.95, 14.26. **ESI-MS: calculated** $[\text{C}_{17}\text{H}_{21}\text{NO}_3 + \text{H}]^+$: 288.1594, **found**: 288.1594.

Ethyl 2-(benzo[d]oxazol-2-yl)-3,3-dimethylbutanoate (S1q)

$^1\text{H NMR}$ (400 MHz, CDCl_3) δ 7.80 – 7.68 (m, 1H), 7.62 – 7.49 (m, 1H), 7.40 – 7.29 (m, 2H), 4.31 – 4.15 (m, 2H), 3.95 (s, 1H), 1.26 (t, $J = 7.1$ Hz, 3H), 1.19 (s, 9H). $^{13}\text{C NMR}$ (100 MHz, CDCl_3) δ 168.74, 162.60, 150.86, 141.04, 125.05, 124.41, 120.26, 110.78, 61.33, 56.48, 35.35, 28.30, 14.27. **ESI-MS: calculated** $[\text{C}_{15}\text{H}_{19}\text{NO}_3 + \text{H}]^+$: 262.1438, **found**: 262.1438.

Ethyl 2-(benzo[d]oxazol-2-yl)-2-phenylacetate (S1r)

$^1\text{H NMR}$ (400 MHz, CDCl_3) δ 7.81 – 7.69 (m, 1H), 7.58 – 7.49 (m, 3H), 7.45 – 7.36 (m, 3H), 7.35 – 7.29 (m, 2H), 5.34 (s, 1H), 4.36 – 4.21 (m, 2H), 1.26 (t, $J = 7.1$ Hz, 3H). $^{13}\text{C NMR}$ (100 MHz, CDCl_3) δ 168.61, 162.42, 151.17, 141.12, 133.61, 129.23, 129.01, 128.56, 125.30, 124.53, 120.44, 110.85, 62.33, 52.46, 14.15. **ESI-MS: calculated** $[\text{C}_{17}\text{H}_{15}\text{NO}_3 + \text{H}]^+$: 282.1125, **found**: 282.1126.

Ethyl 2-(benzo[d]oxazol-2-yl)-2-(thiophen-2-yl)acetate (S1s)

$^1\text{H NMR}$ (400 MHz, CDCl_3) δ 7.78 – 7.71 (m, 1H), 7.56 – 7.48 (m, 1H), 7.38 – 7.30 (m, 3H), 7.24 – 7.20 (m, 1H), 7.05 – 6.99 (m, 1H), 5.63 (s, 1H), 4.28 (q, $J = 7.1$ Hz, 2H), 1.27 (t, $J = 7.1$ Hz, 3H). $^{13}\text{C NMR}$ (100 MHz, CDCl_3) δ 167.70, 161.63, 151.15, 141.04, 134.28, 128.15, 126.95, 126.68, 125.48, 124.65, 120.55, 110.92, 62.71, 47.51, 14.10. **ESI-MS: calculated** $[\text{C}_{15}\text{H}_{13}\text{NO}_3\text{S} + \text{H}]^+$: 288.0689, **found**: 288.0689.

Preparation of thiazole acetate S1m and benzoxazolyl acetates S1aa, S1ab¹⁻³

To a solution of cysteamine (20 mmol, 1.0 equiv.) or 1-amino-2-hydroxybenzene (20 mmol, 1.0 equiv.) and ethyl 3-ethoxy-3-iminopropanoate hydrochloride (20 mmol, 1.0 equiv.) in anhydrous ethanol (40 mL) was stirred for 4 hours at 70 °C. After completion, saturated aqueous NH_4Cl was added and the

aqueous layer was extracted with EtOAc (30 mL × 3). The combined organic layers were washed with brine, dried over Na₂SO₄, filtered and concentrated in vacuo. The crude product was purified by flash chromatography to afford the corresponding product.

To a solution of cesium carbonate (7.5 mmol, 1.5 equiv.) and ethyl 2-(benzo[d]oxazol-2-yl)acetate (5 mmol, 1.0 equiv.) in THF (20 mL) was added alkyl bromide or alkyl iodide (7.5 mmol, 1.5 equiv.) dropwise. The mixture was stirred at 25 °C or 60 °C until the reaction was complete (monitored by TLC). The mixture was quenched by adding H₂O at 0 °C and the aqueous layer was extracted with EtOAc (30 mL × 3). The combined organic layers were washed with brine, dried over Na₂SO₄, filtered and concentrated in vacuo. The crude product was purified by flash chromatography to afford the corresponding product **S1m**, **S1aa**, **S1ab**.

Ethyl 2-(4,5-dihydrothiazol-2-yl)pent-4-enoate (**S1m**)

¹H NMR (400 MHz, CDCl₃) δ 5.89 – 5.71 (m, 1H), 5.14 (dd, *J* = 17.1, 1.3 Hz, 1H), 5.02 (dd, *J* = 17.2, 1.7 Hz, 1H), 4.24 (t, *J* = 8.5 Hz, 2H), 4.21 (q, *J* = 7.1 Hz, 1H), 3.66 (t, *J* = 7.6 Hz, 1H), 3.31 (t, *J* = 8.4 Hz, 2H), 2.80 – 2.68 (m, 1H), 2.66 – 2.54 (m, 1H), 1.27 (t, *J* = 7.0 Hz, 3H). ¹³C NMR (100 MHz, CDCl₃) δ 170.30, 168.28, 134.29, 117.68, 64.36, 59.28, 50.75, 34.96, 34.01, 14.26. **ESI-MS: calculated** [C₁₀H₁₅NO₂S + H]⁺: 214.0896, **found**: 214.0896.

Ethyl 2-(5-(tert-butyl)benzo[d]oxazol-2-yl)propanoate (**S1aa**)

¹H NMR (600 MHz, CDCl₃) δ 7.76 – 7.73 (m, 1H), 7.45 – 7.36 (m, 2H), 4.28 – 4.17 (m, 2H), 4.10 (q, *J* = 7.3 Hz, 1H), 1.71 (d, *J* = 7.3 Hz, 3H), 1.37 (s, 9H), 1.24 (t, *J* = 7.1 Hz, 3H). ¹³C NMR (150 MHz, CDCl₃) δ 170.51, 164.31, 149.05, 148.03, 141.01, 122.85, 116.70, 109.86, 61.88, 40.77, 35.02, 31.88, 15.19, 14.20. **ESI-MS: calculated** [C₁₆H₂₁NO₃ + H]⁺: 276.1594, **found**: 276.1601.

Ethyl 2-(5-methoxybenzo[d]oxazol-2-yl)propanoate (**S1ab**)

¹H NMR (600 MHz, CDCl₃) δ 7.37 (d, *J* = 8.9 Hz, 1H), 7.19 (d, *J* = 2.5 Hz, 1H), 6.91 (dd, *J* = 8.9, 2.6 Hz, 1H), 4.20 (qd, *J* = 7.1, 2.5 Hz, 2H), 4.08 (q, *J* = 7.3 Hz, 1H), 3.83 (s, 3H), 1.70 (d, *J* = 7.3 Hz, 3H), 1.23 (t, *J* = 7.1 Hz, 3H). ¹³C NMR (150 MHz, CDCl₃) δ 170.47, 164.91, 157.30, 145.65, 141.92, 113.68, 110.84, 103.14, 61.88, 56.03, 40.79, 15.13, 14.17. **ESI-MS: calculated** [C₁₃H₁₅NO₄ + H]⁺: 250.1074, **found**: 250.1078.

Preparation of oxazole acetate **S1n**

To a solution of ethyl 2-cyanopropanoate (10 mmol, 1.0 equiv.) and monoethanolamine (10 mmol, 1.0 equiv.) in PhCl (10 mL) was added Zn(OAc)₂ (0.4 mmol, 0.04 equiv.) at room temperature under N₂ atmosphere. The mixture was stirred vigorously and heated at 130 °C for 24 h. After cooling to rt, saturated aqueous NH₄Cl was added to the mixture and allowed to stir overnight. The organic phase was extracted twice with EtOAc. The combined organic layers were washed with brine, dried over Na₂SO₄, and concentrated in vacuo. The crude products were purified by flash column chromatography (petroleum ether/EtOAc) to afford oxazole acetates **S1n** (212 mg, 1.23 mmol, 12% yield).

Ethyl 2-(4,5-dihydrooxazol-2-yl)propanoate (S1n)

ESI-MS: calculated [C₈H₁₃NO₃ + H]⁺: 194.0788, found: 194.0798.

Preparation of benzoxazolyl acetates S1i, S1u, S1v and benzimidazole acetate S1j¹⁻³

To a solution of NaHMDS (10 mL, 2 M in THF, 20 mmol, 2.0 equiv.) in dry toluene (20 mL) at 0 °C was added amide or ester (20 mmol, 2.0 equiv.) dropwise. After being stirred at 0 °C for 30 minutes, 2-chlorobenzoxazole (1.54 g, 10 mmol, 1.0 equiv.) was added at 0 °C. The reaction mixture was allowed to warm to room temperature and stirred for 3 hours. After completion, saturated aqueous NH₄Cl was added and the aqueous layer was extracted with EtOAc (20 mL × 3). The combined organic layers were washed with brine, dried over Na₂SO₄, filtered and concentrated in vacuo. The crude product was purified by flash chromatography to afford the corresponding product.

To a solution of cesium carbonate (7.5 mmol, 1.5 equiv.) and ethyl 2-(benzo[d]oxazol-2-yl)acetate (5 mmol, 1.0 equiv.) in THF (20 mL) was added alkyl bromide or alkyl iodide (7.5 mmol, 1.5 equiv.) dropwise. The mixture was stirred at 25 °C or 60 °C until the reaction was complete (monitored by TLC). The mixture was quenched by adding H₂O at 0 °C and the aqueous layer was extracted with EtOAc (30 mL × 3). The combined organic layers were washed with brine, dried over Na₂SO₄, filtered and concentrated in vacuo. The crude product was purified by flash chromatography to afford the corresponding product S1i, S1j, S1u, S1v.

2-phenylpropan-2-yl 2-(benzo[d]oxazol-2-yl)propanoate (S1i)

Tert-butyl 2-(1-methyl-1H-benzo[d]imidazol-2-yl)propanoate (S1j)

Tert-butyl 2-(benzo[d]oxazol-2-yl)pentanoate (S1u)

13.73. **ESI-MS: calculated** [C₁₆H₂₁NO₃ + Na]⁺: 298.1414, **found:** 298.1424.

Tert-butyl 2-(benzo[d]oxazol-2-yl)hexanoate (S1v)

¹H NMR (400 MHz, CDCl₃) δ 7.75 – 7.67 (m, 1H), 7.54 – 7.47 (m, 1H), 7.35 – 7.28 (m, 2H), 3.90 (t, *J* = 7.7 Hz, 1H), 2.27 – 2.06 (m, 2H), 1.43 (s, 9H), 1.40 – 1.25 (m, 4H), 0.89 (t, *J* = 7.0 Hz, 3H). **¹³C NMR (100 MHz, CDCl₃)** δ 169.10, 163.98, 150.98, 141.19, 124.94, 124.32, 120.11, 110.66, 82.24, 47.50, 29.98, 29.57, 28.01, 22.40,

13.91. **ESI-MS: calculated** [C₁₇H₂₃NO₃ + Na]⁺: 312.1570, **found:** 312.1574.

Reply to comments by Reviewer 3

(1) The authors have done an excellent job and have addressed my concerns as well as those of the other reviewers convincingly. The manuscript can be published as it is.

Answer: We appreciate Reviewer 3 for the favorable comments.